# How do inorganic nitrogen processing pathways change quantitatively at daily, seasonal and multi-annual scales in a large agricultural stream?

Jingshui Huang[1,2], Dietrich Borchardt[2], Michael Rode[2]

[1]Chair of Hydrology and River Basin Management, Technical University of Munich, Arcisstrasse 21, 80333 Munich, Germany

[2]Department of Aquatic Ecosystem Analysis, Helmholtz Centre for Environmental Research - UFZ, Brueckstrasse 3a, 39114 Magdeburg, Germany

*Correspondence to*: Jingshui Huang (jingshui.huang@tum.de)

**Abstract.** Large agricultural streams receive excessive inputs of nitrogen. However, quantifying the role of these streams in nitrogen processing remains limited because continuous direct measurements of the interacting and highly time-varying nitrogen processing pathways in larger streams and rivers are very complex. Therefore, we employed a monitoring-driven modelling approach with high-frequency in-situ data and the river water quality model WASP 7.5.2 in the 27.4-km reach of the 6[th] order agricultural stream Lower Bode (central Germany) for a 5-year period (2014-2018). Paired high-frequency sensor data (15-min interval) of discharge, nitrate, dissolved oxygen, and Chlorophyll-a at upstream and downstream stations were used as model boundaries and for setting model constraints. The WASP model simulated 15-minute intervals of discharge, nitrate and dissolved oxygen with Nash-Sutcliffe-Efficiency values higher than 0.9 for calibration and validation, enabling the calculation of gross and net dissolved inorganic nitrogen uptake and pathway rates on a daily, seasonal and multi-annual scale. Results showed daily net uptake rate of dissolved inorganic nitrogen ranged from -17.4 mg N m$^{-2}$ d$^{-1}$ to 553.9 mg N m$^{-2}$ d$^{-1}$. The highest daily net uptake could reach almost 30% of total input loading, which occurred at extreme low flow in summer 2018. The growing season (spring and summer) accounted for 91% of the average net annual uptake of dissolved inorganic nitrogen in the measured period. In spring, both the DIN gross and net uptake were dominated by the phytoplankton uptake pathway. In summer, benthic algae assimilation dominated the gross uptake of dissolved inorganic nitrogen. Conversely, the reach became a net source of dissolved inorganic nitrogen with negative daily net uptake values in autumn and winter, mainly because the release from benthic algae surpassed uptake processes. Over the five years, average gross and net uptake rates of dissolved inorganic nitrogen were 124.1 and 56.8 mg N m$^{-2}$ d$^{-1}$, which accounted for only 2.7% and 1.2% of the total loadings in the Lower Bode, respectively. 5-year average gross DIN uptake decreased from assimilation by benthic algae through assimilation by phytoplankton to denitrification. Our study highlights the value of combining river water quality modelling with high-frequency data in obtaining reliable budget of instream dissolved inorganic nitrogen processing, which facilitates our ability to manage nitrogen in aquatic systems. This study provides a

methodology that can be applied to any large stream to quantify nitrogen processing pathway dynamics and complete our understanding of nitrogen cycling.

## 1 Introduction

The instream processing of dissolved inorganic nitrogen (DIN) consists of complex and multiple simultaneous pathways (Hensley and Cohen, 2020). The dominant pathway processes include nitrification, denitrification, autotrophic uptake and release, heterotrophic uptake and release, and mineralization (Burgin and Hamilton, 2007; Tank et al., 2018). Disentangling these interacting processes at the reach scale is challenging because they coincide in streams and share the same DIN constituents for their substrates and products. The classic method for reach-scale DIN pathway quantification is the addition of DIN isotope tracers (Mulholland et al., 2008). Using this methodology, Mulholland et al. (2008) quantified the shares of denitrification and assimilation on total nitrate ($NO_3^-$) in-stream uptake in the Lotic Intersite Nitrogen eXperiment (LINX) Project in a wide range of biomes. This method's main limitation is that it is difficult to apply in high-order streams and can only provide snapshots of the highly dynamic processes. For high-order streams, Heffernan and Cohen (2010) developed a method of calculating mass balance using high-frequency measurements to allow partitioning $NO_3^-$ uptake rates into assimilatory and dissimilatory pathways in a spring-fed river. To achieve a continuous estimation of assimilatory uptake, Rode et al. (2016) correlated daily assimilatory $NO_3^-$ uptake with gross primary production (GPP) using high-frequency oxygen and $NO_3^-$ data. However, both methods rely on a robust diel pattern of $NO_3^-$ concentration fluctuations, which is possibly only obtained where external inputs are well constrained and difficult to obtain in large agricultural streams (Hensley and Cohen, 2020). Due to the lack of effective methods, quantification of DIN processing pathways in large agricultural streams remains poorly explored.

Besides, each pathway process is highly time-varying (Hensley and Cohen, 2020). As a vital DIN processing pathway, autotrophic assimilatory uptake is affected by light availability, temperature, autotrophic biota and presents significant seasonal changes (Tank et al., 2017). Furthermore, denitrification, nitrification, and mineralization also exhibit seasonal changes due to the influences of temperature, substrate concentrations, and hydrological conditions (Burgin and Hamilton, 2007; Verstraete and Focht, 1977). Phytoplankton and benthic algae might co-exist in high-order agricultural streams and their metabolism both determine autotrophic assimilatory uptake (Durand et al., 2011; Desmet et al., 2011; Minaudo et al., 2021). The temporal changes in their relative importance for reach-scale assimilatory DIN uptake is still not well known (Jäger and Borchardt, 2018; Riis et al., 2012). Many studies have reported that DIN can be largely assimilated and fixed in benthic algae's biomass during the growing season (Mulholland et al., 2008). However, few studies have paid attention to the fate of assimilated nitrogen (N) during the non-growing season (Riis et al., 2012; Tank et al., 2018). It is unclear whether the DIN release can exceed gross uptake in the non-growing season and if temporally decoupled DIN uptake and release can cause a seasonal shift of DIN release from stream biomass. Von Schiller et al. (2015) quantified N release by comparing gross and net uptake and found that the streams remained at a biogeochemical steady state on a time scale of hours.

However, the knowledge about the DIN release on longer time scales, such as of seasons, still lacks due to measurement constraints (von Schiller et al., 2015). In addition, considering the highly dynamic nature of the processes, it is uncertain to interpolate and aggregate instream DIN uptake rates from snapshot experiment results in the growing season to estimate how effective a river reach is in processing DIN at annual or multi-annual scales. In order to gain overarching insights into the DIN fate in streams, it is therefore needed to quantify the DIN pathways not only on a daily or weekly basis within the growing season but also continuously throughout the hydro-climatic year and beyond.

Instream water quality models offer synthesis tools to study spatio-temporal DIN variation and turnover processes in streams/rivers (Wagenschein and Rode, 2008; Raimonet et al., 2015; Huang et al., 2019). However, the infrequent temporal resolution of monitoring data for model input and testing commonly restricts the reliability of modelling instream DIN processing (Khorashadi Zadeh et al., 2019). A limited temporal resolution of data can result in the equifinality of model parameter sets and considerable model uncertainties due to the insufficient ability to validate single internal reaction processes (Khorashadi Zadeh et al., 2019; Huang et al., 2021). Emerging high-frequency monitoring techniques can provide higher temporal resolution of boundary conditions and support robust calibration and validation of "data-hungry" mechanistic models (Hamilton et al., 2015; Minaudo et al., 2018; Pathak et al., 2021). Thus, combining emerging high-frequency monitoring techniques and river water quality modelling may allow continuous quantification of instream DIN processing pathways and increase their reliability (Rode et al. 2016b).

Therefore, in this study DIN processing pathways were examined using a five-year paired high-frequency water quality dataset from a 27.4-km reach of the 6[th] order agricultural Bode stream in Central Germany and a complementary setup of the river water quality model WASP7.5.2. With this methodological approach, the instream rates of gross and net DIN uptake, denitrification, assimilatory uptake and release by phytoplankton and benthic algae were quantified at daily, seasonal, and yearly scales. The objectives of this study are to answer three questions: (1) how temporally variable are the DIN processing pathways at daily scale? (2) Are there seasonal shifts in the role of river reach as N source and sink? If so, what is the pathway dominating this shift? (3) Based on the continuous multi-year DIN budget, how effective is the river reach in processing DIN and what are its dominant pathways? Finally, our findings highlight the utility and value of high-frequency data to support river water quality modelling in allowing continuous quantification of instream N uptake pathways and thus improving our understandings of the biogeochemical functioning of the large streams on N cycling and fate.

## 2 Material and Methods

### 2.1 Study site

The Bode River is a 169-km long 6[th] order stream in Saxony-Anhalt, originating in the Harz Mountains and discharging into the Saale. It drains a watershed area of 3270 km². The Bode Catchment is one of the most intensively monitored areas within the Terrestrial Environmental Observatories (TERENO) network (http://www.tereno.net) operated by the Helmholtz Centre for Environmental Research (UFZ). The land use within the basin varies along a longitudinal gradient, with the headwaters

dominated by forests while the lower reaches are dominated by intensive agriculture and partly by urban land. Due to intensive agricultural activities, the Bode River is characterized by high $NO_3^-$ concentrations.

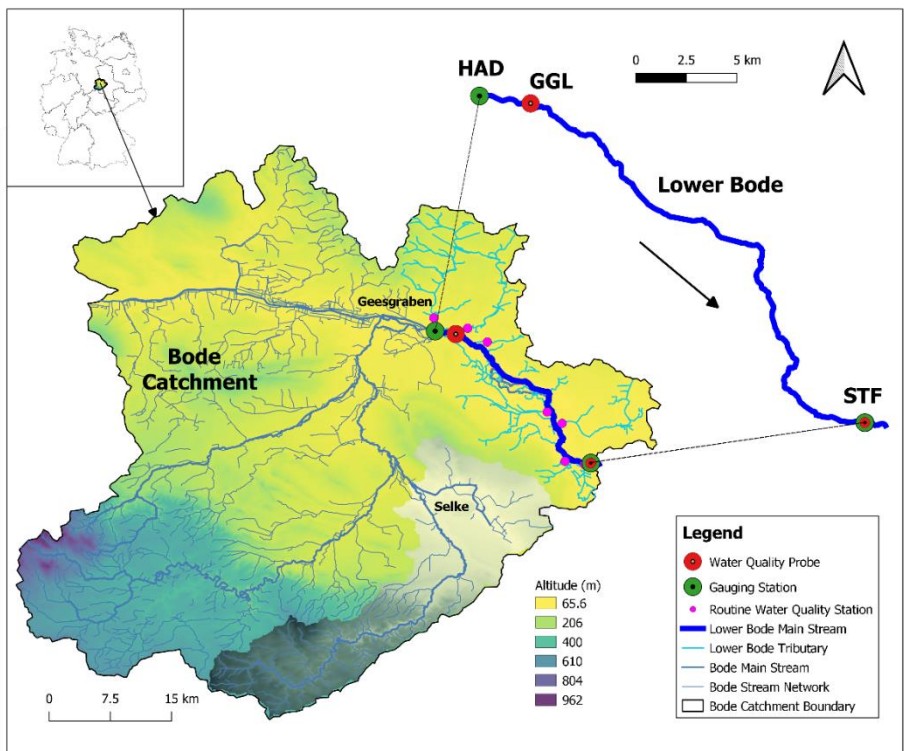

**Figure 1. Bode catchment, stream networks, DEM, study reach of Lower Bode, and site description of Lower Bode. The red circles represent the high-frequency water quality sensors. The green circles denote the gauge stations. The small pink circles depict the**
100 **routine water quality monitoring stations of the tributaries. HAD, GGL and STF stand for the monitoring station names of Hadmersleben, Groß Germersleben, and Staßfurt, respectively. The grey shaded area represents the Selke sub-catchment.**

We studied a 27.4-km reach of the Lower Bode River between Groß Germersleben (GGL) and Staßfurt (STF), with two discharge gauging stations at both ends (Fig. 1). The land use is mainly agricultural and to a minor degree urban. The mean stream width is 20 m. The mean river depth of the reach is approximately 1 m. In the summer low flow period, the river
depth is about 0.5 m, and during the high flow period in winter it can be up to 2-3 meters high. The stream bed substrate is sand and small gravel. The reach has a mean river slope of 0.4 m/km with rectangular or trapezoidal cross-sections. Three weirs are installed in the reach. The original meandering reach has been mostly straightened or re-routed artificially over the past century (LHW, 2012). Lined deciduous trees on the banks only partly shade the reach. The insufficient shading and open canopy allow high irradiance at the water surface and the subsequent development of phytoplankton and benthic algae.
Note that benthic algae refer to the whole primary producer community in benthic habitats including periphyton and macrophytes.

## 2.2 Hydrological and water quality data

Discharge (Q) data at 15-min intervals were obtained at the gauging station Hadmersleben (HAD), which is located 2.7 km upstream of the water quality station GGL, and at the downstream station STF from the hydrological state authority of Saxony-Anhalt (LHW). We assume that discharge at station HAD is also valid for station GGL because no lateral flow contributes to the reach between the two stations.

In the Lower Bode, lateral inputs were not significant compared to the magnitude of the inflow from the main stem (Fig. S1). Daily discharges of the 8 small tributary streams within the study reach were calculated based on monthly discharge measurements from LHW and by the specific discharge method. The reference gauging station measures daily Q at the outlet of a similar stream, Geesgraben, about 0.5 km upstream of HAD (Fig. 1). Detailed descriptions about the Q estimations of the streams are provided in Supplementary Information (SI) Text 1. The sum of the tributary flows accounted only for 5.88% of the upstream inflow over the whole study period. The percentage error of flow balance (eq. S2) between GGL and STF was +0.97% (a positive value means that the sum of inflows was higher than the outflow). Targeted on the low flow period, the percentage error of flow balance was only +0.59% for the extreme summer low flow period in 2018. These small errors, both on a multiannual basis and for the extreme summer low flow period, give a direct indication that lateral groundwater exchange with the watercourse in the study reach does not significantly affect the water balance (SI Text 1). Hourly solar radiation was provided by the German Weather Service (DWD). All data were taken for 5 years (01.01.2014 - 31.12.2018).

Paired high-frequency water quality sensors were installed at GGL and STF. Water temperature (WT), dissolved oxygen (DO), pH, and chlorophyll-a (Chl-a) were measured with a YSI 610 multiparameter probe at 15-min intervals. $NO_3^-$ measurements were also conducted at 15-min intervals using a TRIOS ProPS-UV sensor with an optical path length of 10 mm. Self-cleaning of the sensor was done with air pressure before every measurement. Maintenance of all sensors, including manual cleaning and calibration, was done monthly. All high-frequency water quality measurements were matched with Q by date and time except Chl-a. Chl-a was only measured from spring to autumn, typically from April until November. All high-frequency data were screened to eliminate outliers based on Grubb's test using a moving window method for post-processing. Additionally, the sensor data curves were adjusted to the laboratory measurements from the grab samples by linear regression.

Grab samples were collected monthly for the study period at both stations, filtered and analyzed for $NO_3^-$, ammonium ($NH_4^+$), total nitrogen (TN), orthophosphate ($PO_4^{3-}$), total phosphorus (TP), and dissolved organic carbon (DOC) using standard methods (Rode et al., 2016a). Monthly 7-days biological oxygen demand ($BOD_7$) at HAD and STF were obtained from the Saxony-Anhalt Water Service data portal (GLD) (https://gld-sa.dhi-wasy.de/GLD-Portal/) for the study period. The molar carbon to nitrogen ratio (C/N) of the benthic algae in the Lower Bode is available in Kamjunke et al. (2015) with a value of 9.3. In this reach, the dominant taxonomic composition of benthic algae in biofilms was diatom (Kamjunke et al., 2015).

Bi-monthly routine measurements of $NH_4^+$, $PO_4^{3-}$, TP, DO, and $BOD_7$ for the 8 small tributary streams were obtained from GLD. As all the streams are in agricultural areas, we assumed $NO_3^-$ concentrations are likely to be temporally variable and change with Q. Thus, their concentration-discharge (C-Q) relationships were evaluated for all tributary streams with the long-term monitoring data from 1994 to 2016. All C-Q relationships for the tributaries with available data exhibit enrichment responses with positive slopes (Table S1). Daily $NO_3^-$ concentration for each tributary was calculated by its C-Q linear regression.

## 2.3 Water quality modeling

### 2.3.1 Model description

WASP 7.5.2 Advanced Eutrophication Module developed by the United States Environmental Protection Agency is a river water quality model including nitrogen and phosphorus cycling, dissolved oxygen-organic matter interactions, and kinetics of both phytoplankton and benthic algae (Fig. S2) (Wool et al., 2020). Its DIN processing equations characterize denitrification, assimilatory uptake and excretion by phytoplankton, assimilatory uptake and excretion by benthic algae, mineralization, and nitrification (Table 1). WASP 7.5.2 can simulate diurnal dynamics of water quality variables (Wool et al., 2002) and derive its outputs at the resolution of 15-min intervals, allowing full use of high-frequency data.

**Table 1. Areal rates (in g N m$^{-2}$ d$^{-1}$) of DIN-related processes in the Advanced Eutrophication Module of WASP 7.5.2[1].**

| Process | Notation | Areal Rate |
|---|---|---|
| Denitrification | $U_D$ | $k_{dnit}\theta_{dnit}^{T-20}(\dfrac{K_{NO3}}{K_{NO3}+C_{DO}})C_{NO3} \times z$ |
| Assimilatory uptake by phytoplankton | $U_{A,P}$ | $G_p a_{NC} C_{PhyC} \times z$ |
| Excretion by Phytoplankton | $R_P$ | $D_p a_{NC}(1-f_{ON})C_{PhyC} \times z$ |
| Assimilatory uptake by benthic algae | $U_{A,B}$ | $\rho_{mN}\left(\dfrac{C_{NH4}+C_{NO3}}{K_{SNb}+C_{NH4}+C_{NO3}}\right)\left(\dfrac{K_{qN}}{K_{qN}+q_N-q_{0N}}\right)a_b$ |
| Excretion by Benthic Algae | $R_B$ | $k_{Eb20}\theta_{Eb}^{T-20}qN\, a_b(1-f_{ONb})$ |
| Mineralization | $U_{MIN}$ | $k_{min}E_{min}^{T-20}\left(\dfrac{C_{PhyC}}{K_{mpc}+C_{PhyC}}\right)C_{DON} \times z$ |

[1] Note that we used mg N m$^{-2}$ d$^{-1}$ as the unit for areal rates in the results and discussion section, which needs multiplying the rate values calculated in the table by 1000. The parameters used for the calculations are shown in Table S2. $C_{DO}$, $C_{NO3}$, $C_{PhyC}$, $C_{NH4}$, and $C_{DON}$ represent the concentrations of DO, $NO_3^-$, phytoplankton biomass carbon, $NH_4^+$, and DON, in mg/L. $q_N$

represents algal cell N in $mg_N/g_D$, (D refers to detritus) and $a_b$ represents the bottom algal biomass in $g_D/m^2$. $z$ denotes the water depth (m).

### 2.3.2 Model setup

The WASP model was set up for the 27.4-km study reach of the Lower Bode between GGL and STF for 5 years (01.01.2014 - 31.12.2018). The entire reach was divided into 34 model segments with an average length of 806 m. Average segment riverbed morphology was characterized by 413 cross-sectional profiles provided by LHW (LHW, 2012). The upper boundary condition was forced by the 15-min interval Q and monthly $BOD_7$ at HAD, 15-min interval for $NO_3^-$, DO, and Chl-a, and monthly $NH_4^+$, $PO_4^{3-}$, and TP at GGL. The lateral boundary conditions were defined by the daily Q and $NO_3^-$

concentrations, and bi-monthly $NH_4^+$, $PO_4^{3-}$, TP, DO, and $BOD_7$ concentrations of the 8 tributaries. Environmental conditions, including WT and solar radiation, were provided with the data mentioned in Section 2.2. Hydrological and water quality variables were simulated with variable-timesteps smaller than 1 min. All the input data at different temporal frequencies were linearly interpolated to fit the computation time step of the WASP model. Generally, low-frequency data also followed consistent seasonal patterns according to multi-annual records like e.g., $PO_4^{3-}$ in Fig. S8.

### 2.3.3 Sensitivity analysis, calibration & validation

Before calibration, sensitivity analysis (SA) was conducted to screen the parameters significantly influencing model outcomes. 31 parameters related to the DIN processes were defined with uniform distribution within the ranges previously reported (Table S2). The Elementary Effects (EE) method was used, and the analysis was performed using the SAFE toolbox (Pianosi et al., 2015). The objective functions were defined by the Root Mean Square Error (RMSE) coefficients of $NO_3^-$,

$NH_4^+$, DO, and Chl-a. The most sensitive parameters for $NO_3^-$ and $NH_4^+$, and additional parameters sensitive to DO and Chl-a were identified. After SA, the most identifiable parameters were automatically calibrated using the Gauss-Marquardt-Levenberg algorithm with OSTRICH v17.12.19. (Matott, 2017). The ranges of the selected parameters were defined the same as in the SA (Table S2). The objective function was defined as the weighted sum of square error of the four variables ($NH_4^+$, $NO_3^-$, DO, and Chl-a) at STF for the calibration period of 01.01.2014 - 31.12.2015. We assigned the values to the

other less sensitive parameters within the ranges frequently reported (Wool et al., 2002). After calibration, we evaluated the model performance in Q and water quality variables for the validation period of 01.01.2016 - 31.12.2018 using three criteria: Nash-Sutcliffe-Efficiency (NSE) coefficient, Percent Bias (PBias), and RMSE.

We calculated daily gross primary production (GPP, in g O2 $m^{-2}$ $d^{-1}$) with the 15-min interval DO data using the single-station method (Rode et al., 2016a) and compared it with the sum of GPP of phytoplankton ($GPP_p$) and benthic algae ($GPP_b$)

calculated with the WASP 7.5.2 model results:

$$GPP_p = G_p \times C_{PhyC} \times ROC \times z \qquad (1)$$

$$GPP_b = F_{Gb} \div ADC \times ROC \qquad (2)$$

where $G_p$ is phytoplankton growth rate (d$^{-1}$); $C_{PhyC}$ represent the concentration of phytoplankton biomass carbon (mg C L$^{-1}$); $ROC$ represents oxygen to carbon ratio (gO2/gC, Table S2); $z$ denotes the stream depth (m); $F_{Gb}$ is zero-order growth rate for benthic algae (gD m$^{-2}$ d$^{-1}$); $ADC$ represents detritus to carbon ratio (gD/gC, Table S2). This procedure provides process validation besides variable validation, especially for phytoplankton and benthic algae growth parameters.

## 2.4 Quantification of DIN uptake rates

DIN input and output (in kg N d$^{-1}$) were defined as the sum of the NO$_3^-$ and NH$_4^+$ loadings at GGL and tributaries, and the loading at STF, respectively. Areal rates of DIN-related processes (in mg N m$^{-2}$ d$^{-1}$) were calculated for each segment with the equations in Table 1 and were added up over the study river reach. The DIN inputs, output, and process rates were averaged daily, seasonally, and for the whole 5-year period. The seasons start with spring on Mar. 1$^{st}$, summer on Jun. 1$^{st}$, autumn on Sept. 1$^{st}$, and winter on Dec. 1$^{st}$.

The gross DIN uptake rate ($U_{GROSS}$, in mg N m$^{-2}$ d$^{-1}$) was calculated as the sum of denitrification rate ($U_D$), gross assimilatory uptake rate by phytoplankton ($U_{A,P}$), and by benthic algae ($U_{A,B}$):

$$U_{GROSS} = U_D + U_{A,P} + U_{A,B} \tag{3}$$

The net uptake rate was calculated by subtracting the DIN release rates by phytoplankton ($R_P$), benthic algae ($R_B$), and mineralization ($U_{MIN}$) from $U_{GROSS}$:

$$U_{NET} = U_D + U_{A,P} + U_{A,B} - R_P - R_B - U_{MIN} \tag{4}$$

The percentage gross uptake of total input loading ($E_{GROSS}$, in %) and the percentage net uptake ($E_{NET}$, in %) were calculated

$$E_{GROSS} = \frac{U_{GROSS} \times A \times 10^{-6}}{I} \times 100\% \tag{5}$$

$$E_{NET} = \frac{U_{NET} \times A \times 10^{-6}}{I} \times 100\% \tag{6}$$

where $A$ represents the riverbed area of the study reach in m$^2$; $I$ represents the total DIN input in kg d$^{-1}$; $10^{-6}$ is the unit converter from mg d$^{-1}$ to kg d$^{-1}$. The net assimilatory uptake rates by phytoplankton ($U_{NET,A,P}$) and by benthic algae ($U_{NET,A,B}$) were calculated by subtracting release rates from gross uptake rates:

$$U_{NET,A,P} = U_{A,P} - R_P \tag{7}$$

$$U_{NET,A,B} = U_{A,B} - R_B \tag{8}$$

For better comparing the uptake rate results of this study with others, we calculated the NO$_3^-$ gross and net uptake rate ($U_{GROSS,NO3}$ & $U_{NET,NO3}$) and NH$_4^+$ gross and net uptake rate ($U_{GROSS,NH4}$ & $U_{NET,NH4}$) separately (detailed description in SI Text 2).

## 3 Results

### 3.1 Modeling performance and physio-chemical characteristics

The parameter sensitivity ranking results showed that the parameters related to benthic algae metabolism, including $F_{Gb20}$, $\rho_{mN}$, $K_{Lb}$, $K_{hnxb}$, $k_{Db20}$, $K_{qN}$, $k_{Rb20}$, $f_{ONb}$, $k_{Eb20}$ and $\theta_{Gb}$, influenced the goodness-of-fit of measured and simulated $NH_4^+$ and

$NO_3^-$ the most (Fig. S3, Table 2). Following those, the parameters related to denitrification and phytoplankton processes, including $k_{dnit}$ and $k_{Gmax}$, were also screened as the identifiable parameters (Fig. S3). The most sensitive parameters for simulating N dynamics are related to benthic algae and phytoplankton turnover and denitrification.

**Table 2. Identifiable parameters related to DIN processing and their optimized values.**

| Symbol | Kinetic Constant | Units | Value | Range[1] |
|---|---|---|---|---|
| $F_{Gb20}$ | Benthic algae maximum growth rate | gD m$^{-2}$ d$^{-1}$ | 6.5 | 5 – 100 |
| $\rho_{mN}$ | Maximum N uptake rate for benthic algae | mgN/gD-d | 720 | 200 – 2000 |
| $K_{Lb}$ | Light constant for benthic algal growth | Ly d$^{-1}$ | 130 | 50-300 |
| $K_{hnxb}$ | Ammonia preference for benthic algae | mg N L$^{-1}$ | 0.025 | 0.01 – 0.5 |
| $k_{Db20}$ | Benthic algae death rate constant | d$^{-1}$ | 0.02 | 0.001-0.2 |
| $K_{qN}$ | Half saturation uptake constant for benthic algae intracellular N | mgN/gD | 9 | 5 – 20 |
| $k_{Rb20}$ | Benthic algae respiration rate constant | d$^{-1}$ | 0.2 | 0.05 – 0.2 |
| $f_{ONb}$ | Fraction of benthic algae recycled to organic N | -- | 0.21 | 0 – 0.5 |
| $k_{Eb20}$ | Internal nutrient excretion rate constant for benthic algae | d$^{-1}$ | 0.1 | 0.02 – 0.1 |
| $\theta_{Gb}$ | Temperature coefficient for benthic algal growth | -- | 1.08 | 1.05 – 1.1 |
| $k_{dnit}$ | Denitrification rate constant at 20 °C | d$^{-1}$ | 0.15 | 0 – 0.4 |
| $k_{Gmax}$ | Phytoplankton maximum growth rate constant at 20 °C | d$^{-1}$ | 2.5 | 0.5 – 4.0 |

[1] Sources of literature values: Wool et al. (2002) and Martin et al. (2017).

The optimized model parameters using automatic calibration are presented in Table 2. Simulation results of Q, $NO_3^-$, DO, Chl-a, and $PO_4^{3-}$ and the corresponding evaluation criteria for calibration (2 years) and validation periods (3 years) were shown in Fig. 2 and Table 3. The PBias was lower than 5% and NSE was higher than 0.90 between the simulated and measured high-frequency data of Q, $NO_3^-$, and DO for both periods (Table 3). Our Chl-a simulation results captured the timing of blooms, peaks, and decline of phytoplankton (Fig. 2d). The NSE of Chl-a was relatively low partly due to the

leverage effect of missing the extreme high Chl-a concentrations only on some days during the bloom periods on the overall performance criteria. The underestimation of the spring bloom peaks during the validation period is due to the compensation of reproducing the spring peaks using the same set of parameters during the calibration period. The WASP model simulations of $NH_4^+$, $PO_4^{3-}$, and Carbonous BOD (CBOD) also mimicked the seasonality well (Fig. 2e&2f, S4a, and Table 3).

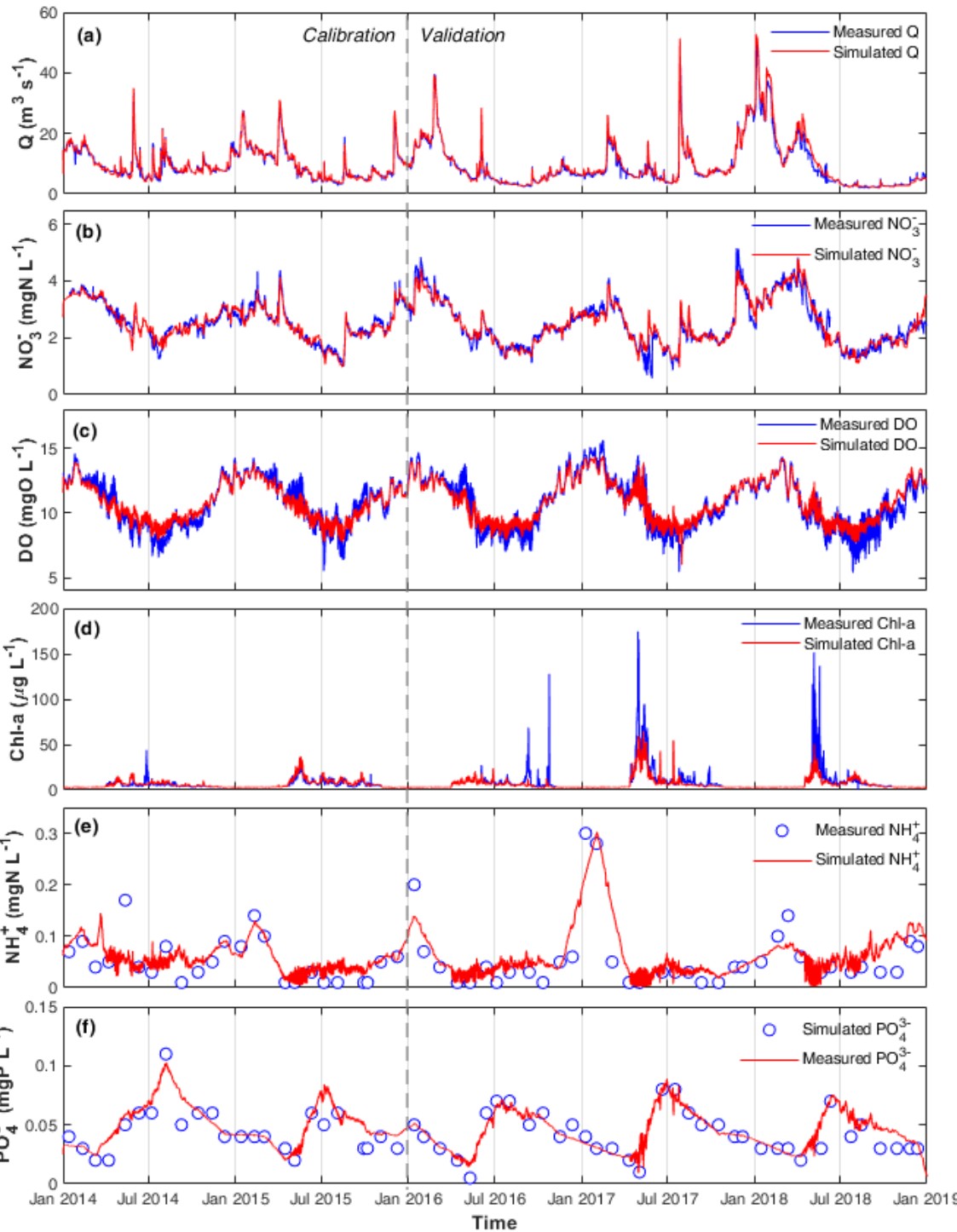

**Figure 2. Measured and simulated (a) Q (b) NO₃⁻ (c) DO (d) Chl-a (e) NH₄⁺ and (f) PO₄³⁻ concentrations in the calibration and validation periods at the STF station.**

**Table 3. Statistics of measurements and model evaluation criteria (NSE, RMSE, and PBIAS) of calibration and validation results at the STF station of Lower Bode River.**

| | Measurement Statistics | | | Calibration (2014-2015) | | | Validation (2016-2018) | | |
|---|---|---|---|---|---|---|---|---|---|
| | Unit | Mean | SD | NSE | PBIAS (%) | RMSE | NSE | PBIAS (%) | RMSE |
| **Q** | $m^3 \, s^{-1}$ | 9.58 | 6.78 | 0.97 | 0.85 | 0.08 | 0.98 | 4.48 | 0.42 |
| **$NO_3^-$** | $mgN \, L^{-1}$ | 2.50 | 0.77 | 0.93 | -0.45 | 0.01 | 0.91 | 1.16 | 0.03 |
| **DO** | $mg \, L^{-1}$ | 10.74 | 1.92 | 0.90 | 0.12 | 0.01 | 0.92 | -0.11 | 0.04 |
| **Chl-a** | $\mu g \, L^{-1}$ | 10.25 | 13.15 | 0.45 | 19.04 | 1.28 | 0.53 | -28.85 | 3.79 |
| **$NH_4^+$** | $mgN \, L^{-1}$ | 0.06 | 0.06 | 0.42 | 16.52 | 0.01 | 0.75 | 22.91 | 0.01 |
| **GPP** | $mgO \, L^{-1}$ | 0.65 | 0.86 | 0.23 | -6.96 | 0.06 | 0.45 | -8.60 | 0.08 |
| **$PO_4^{3-}$** | $mgP \, L^{-1}$ | 0.04 | 0.02 | 0.73 | 11.96 | 0.01 | 0.80 | 6.98 | 0.003 |
| **CBOD** | $mgO \, L^{-1}$ | 4.60 | 0.71 | 0.17 | -13.99 | 0.30 | 0.17 | -14.75 | 0.33 |

A clear seasonal pattern of Q can be observed, with higher values in winter and lower ones in summer (Fig. 2a). Q averaged 9.58 $m^3 \, s^{-1}$, with a minimum of 2.05 $m^3 \, s^{-1}$ during the extreme low flow in August 2018 and a maximum of 49.49 $m^3 \, s^{-1}$ during winter high flow in January 2018 (Table 3, Fig. 2a). $NO_3^-$ concentration was higher in winter when Q was high and lower in summer when Q was low (range 0.57–5.15 and mean 2.50 mg N $L^{-1}$; Fig. 2b). By comparing the concentrations of $NH_4^+$ and $NO_3^-$, we found that $NO_3^-$ was the dominant form of DIN in the Lower Bode, accounting for more than 97% of total DIN (calculated according to their mean values in Table 3). DO concentrations were higher in winter and lower in summer (range 5.40-15.63 and mean 10.74 mg $L^{-1}$; Fig. 2c). Oxygen daily amplitudes were most pronounced during the spring phytoplankton bloom period and the summer low-flow period. In the extreme summer low-flow period of 2018 (validation period), simulated DO amplitudes were smaller than the measured ones (Fig. S5), indicating an underestimation of primary production. Phytoplankton blooms were observed in spring between April and late May, especially in 2017 and 2018, with peak concentrations of Chl-a higher than 150 $\mu g \, L^{-1}$ (Fig. 2e). A good fit of simulated and observed diurnal pattern of Chl-a concentration were observed for the summer low flow (Fig. S5). $NH_4^+$ and $PO_4^{3-}$ concentrations reached the lowest value during the spring blooms. $PO_4^{3-}$ concentration recovered to the highest value usually in summer, while $NH_4^+$ reached the high level in winter. Note that the diurnal patterns of $NH_4^+$ and $PO_4^{3-}$ were observed in the simulation results with a minimum in the afternoon and a maximum around dawn. However, their diurnal variations were smaller than the seasonal ones (Fig. S5).

Simulated GPP from WASP reflected those calculated by the single station method (Fig. 3, Table 3). GPP showed two seasonal peaks each year, one in spring and one in summer (Fig. 3a). The first peak in spring corresponded to the peak of phytoplankton Chl-a concentrations (Fig. 2d), while the other corresponded to the peak of benthic algae biomass (Fig. S4b). Accordingly, the model results showed that GPP was dominated by phytoplankton growth in spring, while in summer, GPP was controlled by benthic algae (Fig. 3b).

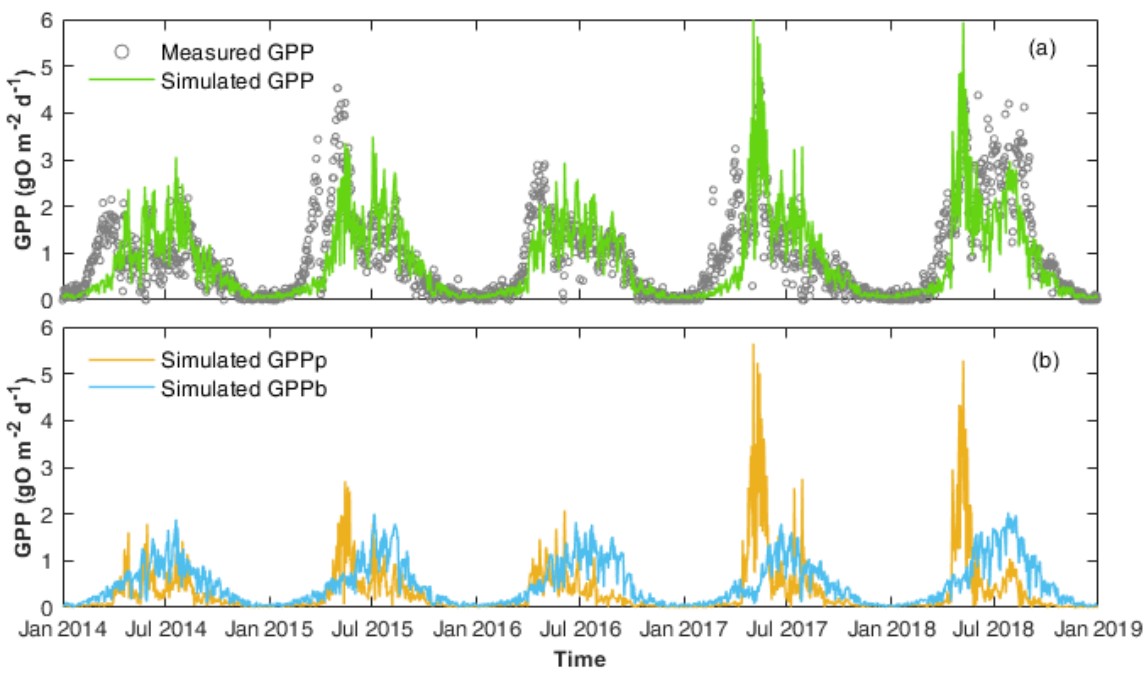

**Figure 3. (a) Comparison of the daily GPP calculated by WASP (simulated GPP) and by the single-station method (measured GPP); (b) Two components of simulated GPP, i.e., GPP by phytoplankton (GPP$_p$) and by benthic algae (GPP$_b$).**

### 3.2 Daily DIN uptake rates

Daily DIN gross uptake rate ($U_{GROSS}$) averaged 124.1 and ranged from 7.9 to 707.9 mg N m$^{-2}$ d$^{-1}$ (Table S3). Daily net uptake rate ($U_{NET}$) averaged 56.8 and ranged from -17.4 to 553.9 mg N m$^{-2}$ d$^{-1}$. Contrary to the seasonal trends of NO$_3^-$ and NH$_4^+$ concentrations, higher $U_{GROSS}$ and $U_{NET}$ were primarily found in spring and summer, whereas the rates were lower in autumn and winter, even with negative $U_{NET}$ (Fig. 4a & 4b). The two highest peak values appeared in the spring of 2017 and 2018 when phytoplankton bloomed and the Chl-a level peaked (Fig. 4a & 4b). The $U_{GROSS}$ and $U_{NET}$ increased significantly with the bloom formations and dropped with their vanishing. In summer, $U_{NET}$ was high most of the time until it decreased towards September (Fig. 4b). The negative values of $U_{NET}$ in autumn and winter showed that the river reach released net DIN. Consistent with the uptake rate, $E_{NET}$ was also higher in spring and summer and lower in autumn and winter. However, the efficiencies in summer were generally even higher than those in spring because of the lower DIN loading in summer (Fig. 4c). Percentage daily net uptake ($E_{NET}$) peaked at 29.1% during the extreme summer low flow of August 2018 (Fig. 4c). The variability of daily net uptake rate by phytoplankton ($U_{NET,A,P}$) was more extensive than that of benthic algae ($U_{NET,A,B}$) (Fig. 4e & 4f). We observed constant negative $U_{NET,A,B}$ in autumn and winter when $R_B$ was higher than $U_{A,B}$. $U_{NET,A,P}$ was also negative but closer to zero compared with $U_{NET,A,B}$ in autumn and winter. Daily denitrification rate ($U_D$), as the only N removal process, averaged 14.1 mg N m$^{-2}$ d$^{-1}$ and ranged from 0.2 to 117.1 mg N m$^{-2}$ d$^{-1}$ (Fig. 4d).

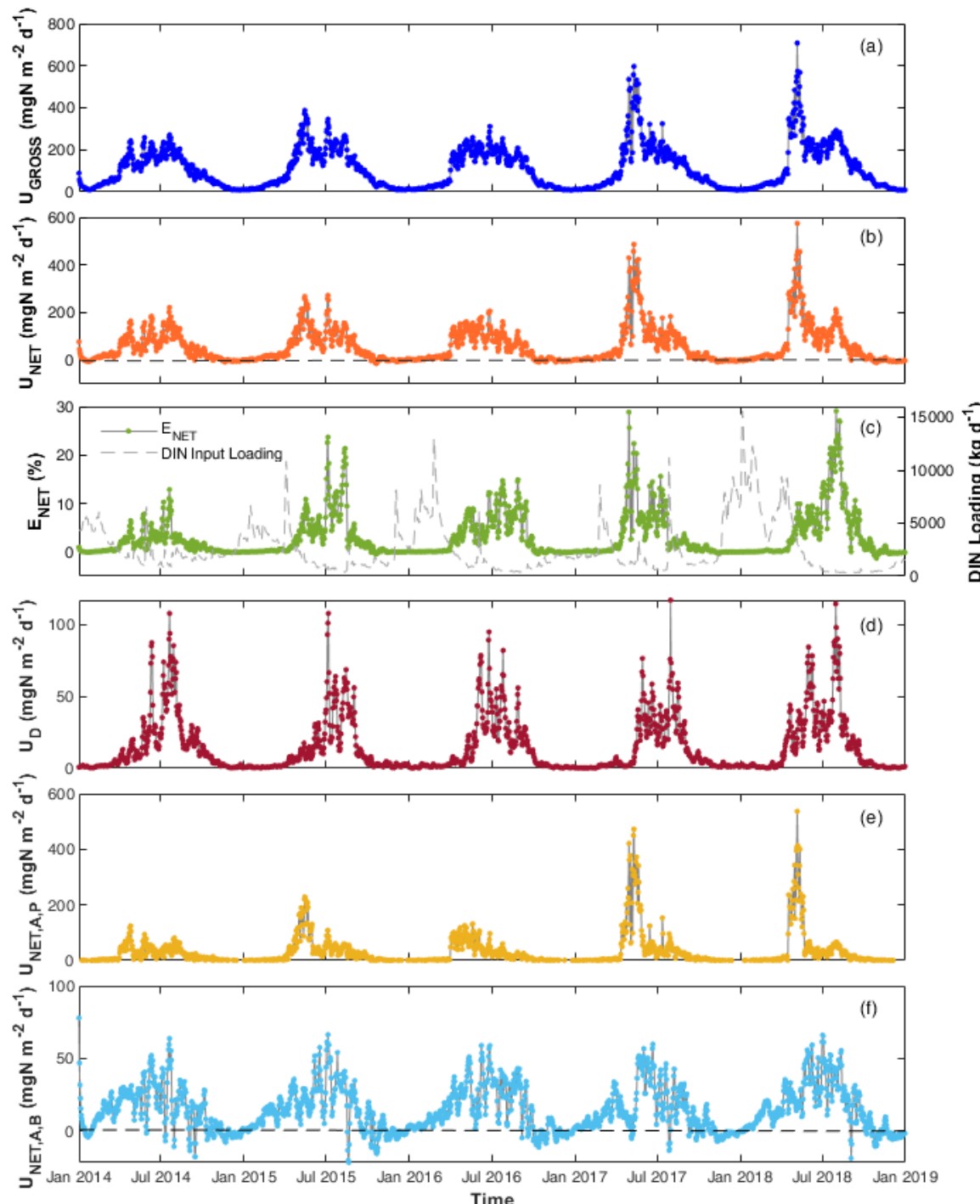

**Figure 4.** Daily DIN (a) gross uptake rate ($U_{GROSS}$), (b) net uptake rate ($U_{NET}$), (c) percentage net uptake ($E_{NET}$), (d) denitrification rate ($U_D$), (e) net uptake rate by phytoplankton ($U_{NET,A,P}$), and (f) by benthic algae ($U_{NET,A,B}$) in the Lower Bode River (2014-2018). Black dash lines in (b) and (f) are zero lines to better show that net N uptake is often negative in autumn and winter.

### 3.3 Seasonal and multi-annual pattern of DIN uptake

DIN fluxes exhibited evident seasonal variation with the highest loadings in winter, accounting for more than half of the total fluxes of the year (Table 4). The seasonal rankings of both averaged DIN gross and net uptake rates were: summer > spring > autumn > winter (Table 4). In spring, both the DIN gross and net uptake were dominated by the phytoplankton uptake pathway. In summer, benthic algae assimilation dominated the gross DIN uptake. However, the contribution of benthic algae to net uptake was limited due to the elevated excretion rate by higher temperatures. Besides, the denitrification rate increased

significantly in summer, accounting for 16.7% of the gross uptake and 37.5% of the net uptake in summer. Thus, the three DIN processing pathways contributed similar shares of DIN net uptake in summer. It should be noted that the averaged net DIN uptake by benthic algae was found to be positive in autumn and winter despite many negative daily values. One explanation is that the net uptake was calculated only for DIN. In the WASP model, algae uptake uses two forms of N source, i.e., $NH_4^+$ and $NO_3^-$, and they excrete in the two forms of DON and $NH_4^+$. The organic fraction of excreted N ($f_{ONb}$,

Table 2) for benthic algae was set to 0.21 in the model after calibration. From the perspective of reactive N (DIN+DON, $N_r$), benthic algae in fact played the role of $N_r$ source in autumn (48.6/(1-0.21) > 52.3; Table 4)).

    5-Year average $U_{GROSS}$ and $U_{NET}$ were 124.1 and 56.8 mg N $m^{-2}$ $d^{-1}$, respectively. On a 5-year basis, gross uptake accounted for 2.7% of total DIN input loading, and net uptake accounted for 1.2% of total loading of the study reach in the Lower Bode. $U_{GROSS}$ via different pathways ranked as assimilatory uptake by benthic algae (72.6 mg N $m^{-2}$ $d^{-1}$) >

assimilatory uptake by phytoplankton (37.2 mg N $m^{-2}$ $d^{-1}$) > denitrification (14.3 mg N $m^{-2}$ $d^{-1}$). In contrast, $U_{NET}$ ranked as phytoplankton net uptake (28.5 mg N $m^{-2}$ $d^{-1}$) > benthic algae net uptake (14.6 mg N $m^{-2}$ $d^{-1}$) > denitrification (14.3 mg N $m^{-2}$ $d^{-1}$).

**Table 4. Seasonal DIN budget including inputs, outputs, and process rates in the Lower Bode (average results from 2014 to 2018).**

| | Spring | Summer | Autumn | Winter | 5-Y Mean |
|---|---|---|---|---|---|
| DIN Inputs (kg N $d^{-1}$) | | | | | |
|   GGL | 2681 | 1117 | 1178 | 5493 | 2617 |
|   Tributaries | 384 | 115 | 117 | 518 | 284 |
|   $\Sigma$ Inputs | 3065 | 1232 | 1295 | 6011 | 2901 |
| DIN Outputs (kg $d^{-1}$) | | | | | |
|   STF | 3026 | 1182 | 1273 | 5999 | 2870 |
| DIN Process Rates (mg N $m^{-2}$ $d^{-1}$) | | | | | |
|   Denitrification | 9.9 | 39.0 | 6.8 | 1.4 | 14.3 |
|   Assimilatory uptake by phytoplankton | 89.4 | 46.0 | 9.8 | 3.8 | 37.2 |
|   Assimilatory uptake by benthic algae | 72.7 | 149.0 | 52.3 | 16.6 | 72.6 |
|   Excretion by phytoplankton | 16.5 | 9.5 | 4.4 | 4.4 | 8.7 |
|   Excretion by benthic algae | 52.3 | 118.7 | 48.6 | 12.5 | 58.0 |
|   Mineralization | 0.2 | 1.7 | 0.4 | 0.03 | 0.6 |

| | | | | | |
|---|---|---|---|---|---|
| DIN gross uptake | 172.0 | 234.0 | 68.9 | 21.7 | 124.1 |
| DIN net uptake | 103.0 | 104.0 | 15.5 | 4.8 | 56.8 |

## 4 Discussion

### 4.1 Seasonal role shift and multi-annual performance of instream DIN processing

The DIN gross and net uptake rates in the Lower Bode showed large seasonal variability, with 85% of gross and 91% of net uptake occurring in the growing seasons, i.e., spring and summer. Thus, the uptake during the growing season dominates the annual uptake amount. $U_{NET,NO3}$ in the growing season in the Lower Bode are comparable to those measured in rivers of similar size from other studies, e.g., Weiße Elster in Germany (Kunz et al., 2017) and 6 subtropical streams in Florida (Hensley et al., 2014) (more comparison in SI Text 3).

Starting from autumn in October, net releases began to appear, and the DIN net release rate could reach 17.4 mg N m$^{-2}$ d$^{-1}$. The net release phenomenon can last until January of the following year. It is worth noting that the net release phenomenon was rarely reported in previous studies (von Schiller et al., 2015). One important reason is most previous measurements were taken by snapshot experiments, which were mainly conducted in the growing season with high biological activity (Mulholland et al., 2008). Still, knowing the net release rate and timing of the shift is also essential. As the net release occurs in the Lower Bode only in autumn and winter, they may not boost downstream ecosystem productivity as the light availability, temperatures and residence time (Fig. S4c) are not favourable in these seasons. The reliable estimation of instream DIN release and timing of the functional shift from net uptake to release can benefit from continuous high-frequency measurements.

The net release is caused by the DIN excretion by primary producers (mainly benthic algae) overpassing the denitrification and assimilatory uptake. Excretion occurs the whole growing season, but it is more than balanced by denitrification and assimilatory uptake. In autumn, the two removal processes are losing importance which leads to a net release. It is challenging to observe the net release from a short-term observation (hours) because N has a "round-trip ticket" to the benthic algae, and the net N uptake can be a quasi-equilibrium (von Schiller et al., 2015). Estimates of the flux of immobilized N back to the water column in the non-growing season have been overlooked for a long time (von Schiller et al., 2015; Riis et al., 2012; Tank et al., 2018). Still, it is critical for our understanding of the entire instream N processing. We overvalue the role of streams in N uptake if we don't consider the N release phenomena.

Analyzing the seasonality of DIN uptake can help us understand the multi-annual performance of the study reach in processing DIN. The 5-year average $E_{NET}$ in the Lower Bode was only 1.2% despite the highest net uptake percentage being close to 30% in the growing season at the daily scale. By only looking at the results from the experiments in the growing season, we might have an impression that the river is very efficient in removing N generally. However, the input DIN loadings of the growing season (spring + summer) accounted for only 37% of the total input loadings for the whole year (calculated from Table 4). In the Lower Bode, nitrate concentration is positively correlated with discharge, which can be

observed in many other agriculture streams in Europe (Moatar et al., 2017). The Lower Bode receives higher N input loadings but becomes less efficient in removing N in the non-growing season when hydraulic conditions (Fig. S4c), temperatures, and light were less favourable for instream uptake processes. Moreover, there is also a seasonal shift from net DIN uptake to net DIN release in an annual cycle. This results in its average net DIN uptake percentage of only 1.2% on a multi-annual basis. From this study, we learned that simple aggregation of instream DIN uptake from snapshot experiment results in the growing season might cause significant uncertainties in estimating annual DIN instream uptake.

## 4.2 Benthic algae and phytoplankton uptake pathways

The DIN assimilation by benthic algae accounts for 59% of the annual gross uptake of the Lower Bode. In contrast, the net DIN uptake by phytoplankton dominates the annual net uptake of the Lower Bode (50%) (Table 4). Our results showed that assimilatory DIN uptakes by phytoplankton and benthic algae both play an essential role in the annual DIN uptake budget in the Lower Bode. The 6[th] order Lower Bode belongs to the mid-reach river system with a transitional size between small streams and large rivers. In the 4[th] order stream Selke upstream of Bode (Fig. 1), benthic algae dominate the assimilatory DIN uptake (Rode et al., 2016a). In contrast, in the 8[th] order River Elbe downstream of Bode, phytoplankton dominates the assimilatory DIN uptake (Kamjunke et al., 2021). In the transitional 6[th] order Lower Bode, phytoplankton gains importance in DIN assimilation. Still, both phytoplankton and benthic algae play a critical overlapping role in assimilatory DIN uptakes at this stream order.

The shift of assimilatory DIN uptake from benthic algae-dominated to phytoplankton-dominated across stream orders well demonstrates the River Continuum Concept (RCC). According to the RCC, mid-reach river systems are broader, deeper, and less strongly influenced by dilution. Autochthonous primary production can occur through planktonic (typically stream order 5-7) development (Durand et al., 2011; Vannote et al., 1980). Recently, Yang et al. (2021) provided evidence of a regime shift in algal biomass over stream-order. Our results further supplemented the algal role transition in terms of DIN uptake across the river network.

Despite both important roles for annual assimilatory DIN uptake, phytoplankton and benthic algae had different dominant seasons, determined by their characteristics and interaction. Phytoplankton formed outbreaks in spring, usually May, especially in 2017 and 2018, whereas benthic algae biomass peaks in the midsummer, typically July. The spring phytoplankton peaks are often observed in many large rivers, e.g., the Elbe, Rhine (Hardenbicker et al., 2014), and Danube (Reynolds and Descy, 1996). The formation of the spring bloom is suggested to be related to the seasonal changes in solar irradiation, water temperature, nutrient availability, and flow condition (Reynolds and Descy, 1996). The most decisive factor of the phytoplankton bloom in the Lower Bode is the seeding Chl-a concentration at the upstream site (Fig. S1d). It basically shapes the timing of the start, peaking and disappearance and the extent of the spring phytoplankton bloom for the whole study reach (Fig. S1d). Favourable instream environment conditions can promote the phytoplankton growth within the study reach, further intensifying the spring bloom peak in the downstream of the reach. The dominant taxa of phytoplankton were identified as diatoms in the Lower Bode (Kamjunke et al., 2015). The average phytoplankton Chl-a levels in the Lower

Bode reached the highest when the water temperature was between 10-14 °C (Fig. S6), which fits the optimal growth temperature of diatoms (Chapra, 2008). As the light availability and other conditions were also favourable in spring, phytoplankton further developed within the study reach, consequently assimilating large amounts of DIN. The disappearance of the spring phytoplankton bloom is mainly driven by the decreasing phytoplankton concentration at the upstream boundary.

In contrast with phytoplankton, benthic algae had the highest biomass and DIN assimilatory uptake in July. Similar seasonal peaking time of benthic algae was also found in other rivers (Desmet et al., 2011; Glasaitė and Šatkauskienė, 2013). The development of benthic algae in summer can be explained by the different environmental factors such as increasing temperature, decreasing turbulence, and grazing during the summer months (Glasaitė and Šatkauskienė, 2013, Jäger and Borchardt, 2018). Note that here benthic algae refer to the whole primary producer community in benthic habitats in our model, where we did not distinguish periphyton and macrophytes. Macrophytes are increasingly important in some large rivers in Europe, e.g., Seine, Moselle, Loire, Ebre because of invasive species (Minaudo et al., 2021). However, periphyton is recognized as dominant in the benthic habitats in the study reach, despite some macrophytes to a small partition (LHW, 2002). Although the newest version of WASP8 can describe them separately (Wool et al., 2020), we prioritized avoiding introducing more model parameters by increasing the model structure complexity, but without sufficient data to identify in our modelling practice. More investigations are needed to determine the reason for the seasonal shift of phytoplankton and benthic algae dominance, considering the development of phytoplankton in the course of the whole Bode River.

The differences between phytoplankton and benthic algae in DIN assimilation were also reflected in the characteristic time. The DIN uptake by phytoplankton was 'acute,' while uptake by benthic algae was 'chronic' (day-week vs. month-years) (Lepoint et al., 2004). The phytoplankton biomass can be temporally very dynamic. It can increase rapidly under favourable conditions within days to weeks in the blooming phase assimilating massive DIN within a short period. Due to the high dynamics of phytoplankton, paired high-frequency monitoring data can better define boundaries of seeds and capture instream growth rates, ensuring a high-accuracy estimation of DIN assimilation. Nevertheless, uncertainty still exists in the estimation results, because of the difficulties inherent in the use of the high-frequency chlorophyll fluorescence signal as indirect measures of phytoplankton biomass and lack of direct measurements (Hamilton et al., 2015). Hamilton et al. (2015) shows the impact of non-photochemical quenching on high-frequency Chl-a fluorescence signals in summer for Lake Rotoehu. However, a recent study from Pathak et al. (2021) found no impact of quenching on high-frequency Chl-a measurements in the Thames River and observed diurnal pattern of Chl-a concentration, which can be reproduced by the model. Similarly in our study, the diurnal pattern of the Chl-a signals related to biomass was also found for the summer low flow condition when the Chl-a concentration is at low level (Fig. S5). However, during the spring bloom in 2017 and 2018, the signal of Chl-a sometimes fluctuates wildly within a day, which cannot represent the actual diurnal biomass changes. In contrast, the biomass change of benthic algae is suggested to be gradual and continuous unless disturbances, such as detachment from the river bottom in high flow events (Jäger and Borchardt, 2018; Rimet et al., 2015). As the DIN uptake or release by benthic algae is closely related to the changes in their biomass, more measurements of benthic algae biomass

across seasons are strongly recommended to support the estimation of the assimilatory DIN uptake via benthic algae with more evidence.

The DIN assimilation by phytoplankton and benthic algae showed another significant difference in terms of gross vs. net uptake. On a 5-year average basis, the net DIN uptake by phytoplankton was 77% of its gross uptake in the Lower Bode, while the net DIN uptake by benthic algae was only 20% of its gross uptake (Table 4). The difference is mainly because the phytoplankton can be transported downstream beyond the study reach, while the benthic algae are fixed in the river bottom in the reach. The assimilated DIN by phytoplankton can be transported out of the reach before it has enough residence time to release. Consequently, massive gross and net uptake in the study reach can be observed simultaneously during the spring phytoplankton blooming phase in the Lower Bode. In contrast, the uptake and excretion of N by the benthic algae is happening simultaneously within the study reach as the assimilated N does not move downstream like in phytoplankton. Besides, benthic algae's biomass completes a cycle on a yearly basis. Large amounts of N assimilated and stored in the benthic algae during the growing season will be released back into the waters after they die (Desmet et al., 2011). In the end, although the gross uptake by benthic algae was higher than by phytoplankton yearly, its net uptake was lower than by phytoplankton at the reach scale (Table 4). Note that our release calculations are conservative estimates because N release in the form of DON was not included and we were not able to measure particulate N release for the stream reach. Specifically, during higher flows, benthic algae and particulate N can be scored from the stream bottom and carried downstream out of our study reach (Jäger and Borchardt, 2018).

Nevertheless, the high DIN net uptake via phytoplankton does not mean DIN is completely removed from the riverine system. Phytoplankton can settle and become a nutrient source further downstream. On the other hand, the low net uptake by benthic algae does not mean that they do not play an essential role in instream N processing. Although the assimilated N by phytoplankton and benthic algae is temporarily stored in their biomass pools and eventually sustains the nutrient spiralling downstream, their uptake contributes to downstream water quality by retarding N downstream transport before denitrification or ultimate burial (Hall et al., 2009; Desmet et al., 2011). Cooper and Cooke (1984) also concluded that the aquatic vegetation modified catchment nitrogen export in terms of form and timing, rather than acting as a net remover from studying nutrient dynamics in a New Zealand stream. This downstream impact is not restricted to N but is also valid for phosphorus (P). Due to the relatively stable stoichiometric N:P relationship in benthic algae (Redfield, 1958; Rutherford et al., 2020), we can assume that nutrient uptake is also valid for P. Therefore, temporal sequestration and associated turnover time delays the downstream transport of nutrients (Ensign and Doyle, 2006) and can lower available downstream P in ecologically relevant spring and summer periods. Our results show that N is strongly released from biomass during less biologically active periods in autumn and early winter, and this is likely also the case for P.

### 4.3 Relevance of P in modelling DIN uptake

Evidence from the model results shows clearly that both phytoplankton and benthic algae in the Lower Bode were P-limited and never reached N limitations (Fig. S7) similar to other agricultural streams/rivers in Europe(Descy et al., 2011; Minaudo

et al., 2018). Especially, P is limiting when the spring bloom happens. Therefore, P is an important aspect in terms of
440 modelling the phytoplankton and benthic algae metabolisms and further the DIN processes and pathways. Although this study has a very good dataset with paired high-frequency measurements for many variables, still some water quality variables, e.g., $PO_4^{3-}$, are not yet available at high frequency. Linear interpolation of $PO_4^{3-}$ concentration was implemented based on the consistent observed seasonal patterns with multi-annual records at GGL and STF (Fig. S8). Nevertheless, there could be discontinuities of $PO_4^{3-}$ concentrations falling out of the linear interpolation line of monthly measurements, which
is most probably caused by storm events. As P availability is a limiting factor for the phytoplankton and benthic algae growth (Fig. S8), such changes can reduce or enhance the algae growth limitation in the short term, thus affecting their growth rate, which in turn affects the DIN assimilatory uptake rate. This effect might be stronger for phytoplankton (Fig. S7) because benthic algae have an internal phosphorus storage in their biomass that can adapt and buffer the response to the sudden change in P availability in the environment.

Our model was able to reproduce the phytoplankton growth between GGL and STF for 5 years, proved with the high-frequency paired Chl-a measurements (Fig. 2d). Hence, we can assume that the impact of $PO_4^{3-}$ on phytoplankton growth was also captured reasonably. This might also be true for benthic algae, as proved by the GPP calculation, which represents the sum of phytoplankton and benthic algae metabolism. Furthermore, we can assume that $PO_4^{3-}$ variation is discharge dependent (r=-0.35, $p<0.05$ at GGL). This means that $PO_4^{3-}$ concentration does not vary much during stable low flow
conditions. The highest phytoplankton growth is mostly associated with stable low flow conditions and the absence of discharge disturbances. Taking this into account it may explain why our infrequent $PO_4^{3-}$ concentration measurements still allowed a reasonable reproduction of phytoplankton growth and DIN uptake calculation.

Besides, $PO_4^{3-}$ concentration is susceptible to diel cycles, which cannot be reproduced by linearly interpolating the monthly measurements. Our modelling results show that diurnal variations of $PO_4^{3-}$ are in a small range, which cannot
interfere with the robust long-term seasonal patterns in $PO_4^{3-}$ data (Fig. S5). In other words, the magnitude of diurnal variations caused by the instream processes does not overshoot the seasonal pattern. Therefore, the uncertainty in estimating reach-scale N uptake by linearly interpolating the monthly data which might contain diel patterns is constrained to a small magnitude on a daily scale. Nevertheless, there might be room to improve phytoplankton simulations and DIN uptake with high-frequency $PO_4^{3-}$ measurements. Ongoing work investigates short-term variation of $PO_4^{3-}$ which offers 30-minute
measurement frequencies to better link $PO_4^{3-}$ and DIN uptake by primary production.

**4.4 The value of using high-frequency data in quantifying N pathways**

This study improved confidence in quantifying the instream N processes using a combination of high-frequency data and a water quality model from several aspects.

First, the synchronous continuous high-frequency $NO_3^-$ and Q data accurately define the upper boundary conditions for
DIN external loadings. Hensley et al. (2019) suggested that DIN fluxes are highly uneven in time, and a significant fraction of yearly export occurs during a few high flow events. Therefore, missing some high flux events may cause a significant

deviation in estimating the DIN uptake in the model, especially on multi-annual scales. As shown in Fig. 5, some concentration peaks caused by storm events were found between the monthly $NO_3^-$ data points.

Second, the GPP from model results can be validated by calculated GPP through high-frequency DO measurements. Recent work suggests that nutrient fluxes are large relative to autotrophic demand in streams that are no longer N-limited (Covino et al., 2018). Thus, validation of simulated GPP is vital to constrain the model parameterization of assimilatory uptake processes. In another study, we have proved that using 15-min interval DO sensor data can improve the identifiability and reduce uncertainties of the parameters for phytoplankton and benthic algae metabolisms using a Bayesian inference approach (Huang et al., 2021). As the autotrophic assimilation plays such an important role in DIN uptake in the Lower Bode, better parameter identification of algae-related processes using high-frequency DO data supports the quantification of DIN uptake processes.

Third, paired high-frequency Chl-a data enabled us to approximate the growth rate of phytoplankton. As shown in Fig. 5, by using routine monitoring frequency data (e.g., monthly data) of Chl-a concentrations one can easily miss its concentration peaks between two monthly measurements. Thus, using monthly Chl-a concertation data for model calibration will cause difficulty and uncertainty in determining the phytoplankton growth and its nitrogen uptake. Besides, high-frequency $NO_3^-$ data are useful in determining how much $NO_3^-$ uptake occurred during the phytoplankton bloom as they can capture short-term but massive uptake-induced reductions in $NO_3^-$ concentration (Fig. 5). Based on the total GPP and phytoplankton GPP, the assimilatory DIN uptakes by phytoplankton and benthic algae are disentangled with more evidence.

Fourth, the downstream high-frequency $NO_3^-$ concentration signal is a synthesis of upstream delivery and internal biogeochemical processing signals. By ascertaining the external loading, downstream $NO_3^-$ concentration signal and internal assimilatory uptake processes, the denitrification rate can be well constrained in the model. Especially in the summer low-flow period, the $NO_3^-$ simulation is sensitive to denitrification because it accounts for over 30% of the net uptake in the Lower Bode. Note that this $NO_3^-$ removal can be dedicated to the sum of denitrification plus heterotrophic uptake. As heterotrophic uptake is not described in the WASP, these two processes cannot be disentangled.

Finally, a continuous estimation of DIN processing via multiple pathways in the Lower Bode was achieved by incorporating high-frequency data into water quality modeling. The significance of our study lies in the overarching picture of the N processing in a 6[th] order agricultural stream, which deepens our systematic insights into the processes and facilitates our ability to manage aquatic systems.

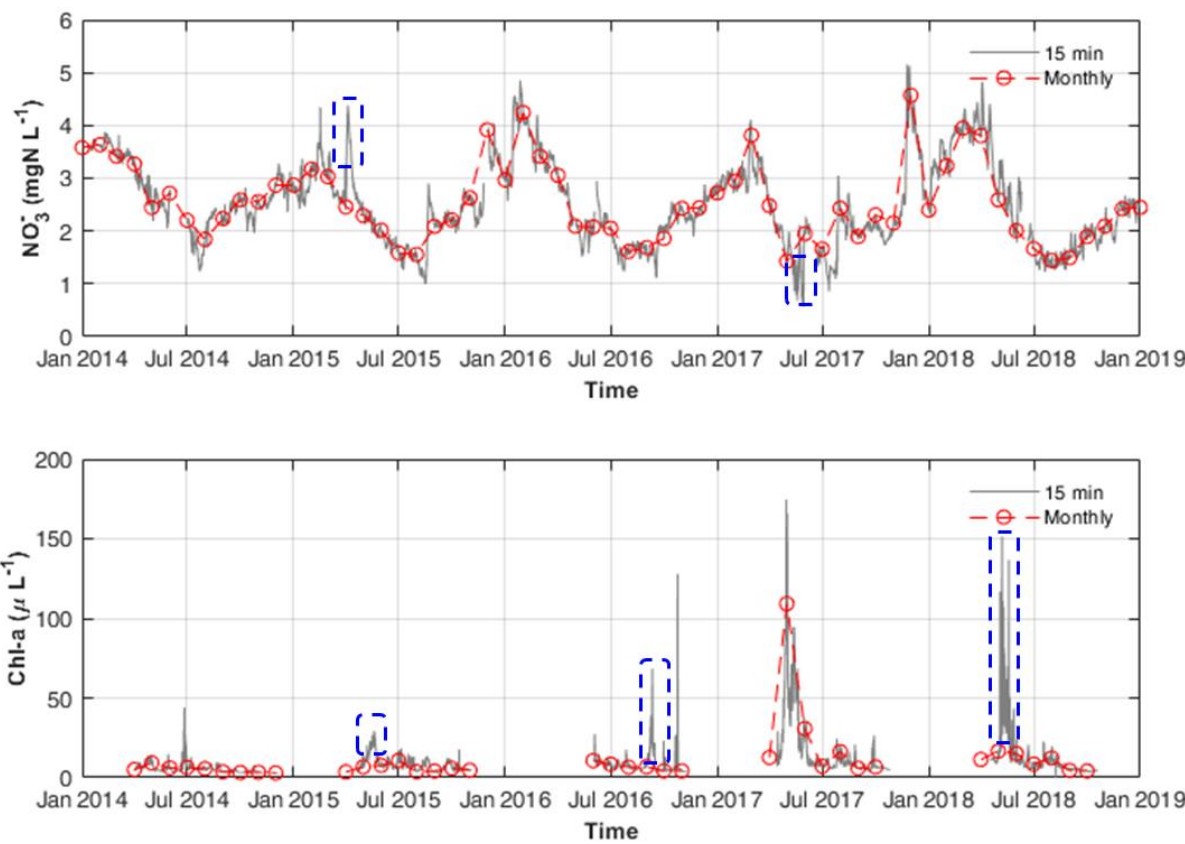

**Figure 5. 15-min interval and re-sampled monthly NO₃⁻ and Chl-a measurements at STF; Blue dashed rectangles illustrate the examples in the discussion Section 4.4.**

## 5 Conclusions

High-frequency data from autonomous sensors offer an opportunity to align observations and simulations from a water quality model. The combination of model and high-frequency data has great potential to deliver the long-needed inference about continuous quantification of instream DIN processing pathways. In this study, we mimicked the patterns of Q, $NO_3^-$,
and DO with the NSE values higher than 0.9 in the Lower Bode using the WASP EUTRO model. This is one of the few model testings with both simulated and measured state variables at such high temporal resolution (15-min interval). On this basis, our model could adequately infer continuous DIN processing, including denitrification, assimilatory uptake and release pathways at daily, seasonal and multi-annual scales and the three research questions could be answered. The daily
DIN net uptake rates were highly temporally variable, ranging from -17.4 mg N m⁻² d⁻¹ to 553.9 mg N m⁻² d⁻¹. Seasonal role shifts of the Lower Bode from N net sink to source were captured and the dominating process causing this shift, namely net N release from benthic algae in the non-growing season, was quantified. Based on the continuous DIN budget across 5 years, the percentage gross uptake in the Lower Bode was only 2.7%, which was dominated by benthic algae assimilation, followed

by phytoplankton assimilation and denitrification. Our results demonstrate the WASP EUTRO model's ability to estimate
DIN processing pathways in a large agricultural lowland stream where benthic algae and phytoplankton co-exist. The long-term high-frequency data increased the reliability of the process estimation from the model through both state and process validations. Nevertheless, in other aquatic systems, the dominant N processing pathways may be different. If the high-frequency measurements happen to have discrepancies with model results in other systems, it also provides an opportunity to identify the mechanisms that may be incompletely represented in the model formulation. In the end, the approach of combining models with high-frequency measurements provides a tool to drive forward more rigorous model assessment and process representation. The resulting improvements in model process representation and performance provide opportunities to move beyond quantifying current snapshot instream N processes and into a domain of dynamic continuous estimation and prediction.

**Data availability**

The high-frequency water quality from the sensors used in this study are archived in the TERENO database and available for the scientific community upon request through the TERENO-Portal (www.tereno.net/ddp). The discharge data are freely available and downloadable from the data portal of the State Office of Flood Protection and Water Quality of Saxony-Anhalt (https://gld-sa.dhi-wasy.de/GLD-Portal/). The water quality data of the tributaries are also downloadable from the same portal. The hourly solar radiation data can be downloaded from open data portal of German Meteorological Service (https://opendata.dwd.de/).

**Author contribution**

JH: conceptualization, methodology, software, validation, formal analysis, investigation, writing- original draft preparation, visualization. DB: resources, writing- reviewing and editing, funding acquisition. MR: data curation, writing- reviewing and editing, supervision, project administration.

**Acknowledgements**

We thank the flood protection and water management agency of the state of Saxony-Anhalt, Germany (LHW), for providing data on water discharge and routine water quality data. We also thank the TERENO (Terrestrial Environmental Observatories) project to support the high-frequency monitoring in Lower Bode and Uwe Kiwel to maintain the sensor measurements. J. Huang would like to thank the financial support from the CSC-DAAD Postdoc Fellowship. We appreciate Dr. Camille Minaudo and an anonymous reviewer for their valuable comments and suggestions in helping us improve the manuscript.

**Competing interests**

The authors declare that they have no conflict of interest.

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
