# Peer review of "How do inorganic nitrogen processing pathways change quantitatively at daily, seasonal and multi-annual scales in a large agricultural stream?"

_Hydrology and Earth System Sciences, 2021_

## Author Comment (AC1)

**Dear Editor and Reviewer,**

Thank you for your letter and the reviewer' comments concerning our manuscript entitled "How do inorganic nitrogen processing pathways change quantitatively at daily, seasonal and multi-annual scales in a large agricultural stream" (hess-2021-615). These comments are all valuable and very helpful for revising and improving our paper, as well as providing significance to our research. We have addressed the comments carefully and made corresponding responses. The responses to the reviewer's comments are listed below.

**Review Report #2**

The study aims to combine high-frequency water quality measurements with a hydrochemical model to improve understanding of dissolved inorganic nitrogen dynamics in a large stream.

The presented results showing how DIN changes from daily to interannual scales together with underlying pathways are convincing and well presented.

**Response:** Thank you very much for the positive general comment.

What is not clear, however, is the benefit of using combined high-frequency data and a hydrochemical model. It seems to me that HF data are purely used to validate model estimates of GPP. If there is an information gain resulting from using this combined approach, it is not clear in the paper. Perhaps contrasting model validation with high- vs low-frequency data would visualize this gain?

If the model can be equally well validated using low-frequency data, what is the benefit of using high-frequency data? This point needs to be clarified by the authors.

**Response:** Thanks a lot for raising the point on the benefit of using high-frequency data. First of all, we have used a paragraph to discuss the value of using high-frequency data in quantifying N pathways in Section 4.3. In our study, high-frequency water quality data have been used as input of the upper boundary condition at GGL and for calibration against the simulation results at STF. There are three high-frequency water quality variables: DO, Chl-a, and $NO_3^-$. Also, low-frequency data can be defined by several different lower frequencies, e.g., daily, weekly, and monthly. Theoretically, we can test the benefits of using high-frequency data as input or for calibration, of each variable, against the different temporal resolutions of low-frequency data in the water quality modelling of the Lower Bode. This will create a lot of combinations of test scenarios, which will constitute a complete individual new study. But this is out of the scope of the current study. Nevertheless, we still wanted to touch on this topic in the paper.

As the reviewer has noticed, we mentioned one very important extra value of using GPP calculated from high-frequency DO data for model calibration and validation. Most important is that high

frequency data of DO are simply a prerequisite to allow GPP and associated N uptake calculations. Moreover, high-frequency data allow increasing the data support for the modelling. In another study, we have proved that using 15-min interval DO sensor data can improve the identifiability and reduce uncertainties of the parameters for phytoplankton and benthic algae metabolisms using a Bayesian inference approach (Huang et al. 2021)[1]. A detailed analysis is subject of ongoing work. So far, the results of Huang et al. (2021) clearly show that the uncertainties in water quality modeling can be reduced by using high-frequency DO data. As the assimilation by algae plays such an important role in N uptake in the Lower Bode, better parameter identification of algae-related processes using high-frequency DO data supports the quantification of N uptake processes. We will include the discussion above and cite our EGU publication as evidence for the benefit of using high-frequency DO data.

Next, also the value of high-frequency $NO_3^-$ and Chl-a data for model calibration can be tested and quantified using a similar approach done by Huang et al. (2021). However, in the current manuscript, the benefits of using high-frequency $NO_3^-$ and Chl-a data in water quality modelling were explained with 3 arguments in Section 4.3 in a descriptive way and we will underpin this with visualisation of some more time series data showing differences between high and low frequency data. We will supplement a figure with both 15-min interval and monthly $NO_3^-$ and Chl-a data at STF and add some discussion elements in Section 4.3 supported by the visualisation of the water quality data at high frequency and low frequency shown in Figure R9.

For supplementing the third argument about the benefit of using high-frequency Chl-a data in Section 4.3, we can see from Figure R9 clearly, that using the monthly data of Chl-a concentrations can easily miss its concentration peaks between two monthly measurements, e.g., spring blooms in 2015, 2016 and 2018. Thus, using Chl-a concertation data at monthly frequency for model calibration will cause difficulty and uncertainty in determining the phytoplankton growth and eventually quantifying its nitrogen uptake. This is, although to some lower extent, also true for $NO_3^-$ data. The peaks between monthly $NO_3^-$ data can be caused by storm events and short-term low values can be caused by instream biogeochemical processes; some examples are marked in Figure R9. These high frequency values are useful in determining how much $NO_3^-$ uptake occurred during the phytoplankton bloom in the water quality model. This suggests that high-frequency Chl-a and $NO_3^-$ data can help quantify the uptake processes. The above illustration will be supplemented to support the discussion in the revised version.

[1] Huang, J., Merchan-Rivera P., Chiogna G., Disse M., and Rode M.: Can high-frequency data enable better parameterization of water quality models and disentangling of DO processes? EGU General Assembly 2021, Online, 13–30 April 2021, EGU2020-18622, https://doi.org/10.5194/egusphere-egu21-8936, 2021.

[Figure]

[Figure]

**Figure R9** 15-min interval and monthly $NO_3^-$ and Chl-a measurements at STF; Cyan dotted lines illustrated the examples in the discussion Section 4.3.

**Minor things:**

Line 10, either stream or river

**Response:** Thanks! We will correct it to stream reach.

Do not use abbreviations in the abstract

**Response:** OK, we will avoid using abbreviations in the abstract of the revised version.

'We assume that discharge at station HAD is also valid for station GGL because no lateral flow contributes to the reach between the two stations' – how can you be sure? It is a long stretch of 2.7 km.

**Response:** We selected the location of the GGL station to guarantee complete mixing of upstream tributaries. Between the HAD and GGL stations, there is no tributary or drainage ditch. In the response to Reviewer 1 we already explained that the water budget is well balanced for the whole 27.4 km study reach between GGL and STF assuming the same discharge at HAD and GGL. Furthermore, calculation using the fully distributed hydrological mHM model indicated that mean yearly groundwater recharge

in the lower Bode valley were always below 10 mm since 2014 (Zhou et al. in review)[2]. Therefore, it is also very unlikely that there is considerable lateral input from the 2.7 km reach. Therefore, even if we cannot completely discard lateral inflow within this 2.7 km because of missing direct measurements, lateral input can be assumed to be negligible in the water balance of the whole reach.

Discussion title Seasonal role shift and multi-annual performance – is not clear

**Response:** Thanks a lot for mentioning this. We will change it to "Seasonal role shift and multi-annual performance of instream N processing."

Line 301 N has a round-trip ticket to - ?

**Response:** The algae biomass. This will be supplemented to the manuscript.

Once again, we appreciate the critical comments and constructive suggestions from the reviewers very much.

Jingshui Huang

Contact author behalf of all co-authors
* * *
[2] Zhou, X., Jomaa, S., Yang, X., Merz, R., Wang, Y., Rode, M.: (in review) Exploring the relations between sequential droughts and stream nitrogen dynamics in central Germany through catchment-scale mechanistic modelling (submitted to J. of Hydrol.)

---

## Author Comment (AC2)

**Dear Editor and Reviewer,**

Thank you for your letter and the reviewer' comments concerning our manuscript entitled "How do inorganic nitrogen processing pathways change quantitatively at daily, seasonal and multi-annual scales in a large agricultural stream" (hess-2021-615). These comments are all valuable and very helpful for revising and improving our paper, as well as providing significance to our research. We have addressed the comments carefully and made corresponding responses. The responses to the reviewer's comments are listed below.

**Review Report #1**

**General Comment**

This manuscript by Huang et al. presents a modelling study where a biogeochemical model is being used to improve our understanding of nitrogen biogeochemical processes along a river reach of the Bode River, Germany. The model is forced and compared with near-continuous measurements in the river main stem. The model performs very well at representing the observations at the downstream station. The very good quality of model outputs enables to identify and quantify the main processes and pathways for DIN in a large lowland river, and focuses in particular on N uptake by primary producers, to show how much of the DIN entering the system is eventually transformed before it exists the river reach considered. This being looked at at short, seasonal and interannual timescales.

The whole study is sound and clear, very well written and organized. The overall quality is excellent, although some elements raised some important questions that I think need to be answered to.

**Response:** Thanks a lot for the positive general comment.

**First,** phytoplankton in the low Bode River is likely phosphorus limited. The whole study focuses on N processes and pathways, but the reason behind needs to be further explained. I was surprised there are no results or data inputs shown for P in the main manuscript (they are shown for $PO_4^{3-}$ in SI), when this is certainly a critical constraint for studying the dynamic of phytoplankton and benthic algal biomasses and their metabolisms. I think these results need to be shown in the main text, and deserves some elements of discussion.

**Response:**

Thanks a lot for addressing the significance of P in the study, which we did not interpret much in the manuscript. We fully agree with the reviewer on that P is an important aspect in terms of studying the phytoplankton and benthic algae metabolisms and further the N processes and pathways. To enhance the process understanding with the relevance of P, we will act in several aspects below.

First, we will move the simulation results of $PO_4^{3-}$ (currently in Figure S3(a) in the SI) to Figure 2 in the main manuscript to show together with the other simulation results. This will help the readers have an overarching view on a) seasonal pattern of the phosphorus, b) model performance on phosphorus simulation, and c) interaction with other water quality constituents, especially for N, Chl-a, etc.

Second, we will increase some elements of discussion related to P. Evidence from the model results proves that the reviewer is correct about that both phytoplankton and benthic algae in the Lower Bode are P-limited in terms of nutrient limitation. The two plots below show the N and P limitation factors calculated in the model for phytoplankton and benthic algae respectively. In our model, the nutrient limitation factor is calculated as the minimum value of the N and P limitation factors. As shown in the figure below, the P-limitation factor values are below N-limitation factor values in the whole simulation period. This means P is the limiting nutrient element for both phytoplankton and benthic algae in the Lower Bode, similar to other agricultural stream/rivers in Europe[1]. Especially, the phosphorus is limiting when the spring bloom happens. This figure will be added in the SI for a basis of discussion.

[Figure]
* * *
[1] Minaudo, C., Curie, F., Jullian, Y., Gassama, N., and Moatar, F.: QUAL-NET, a high temporal-resolution eutrophication model for large hydrographic networks, Biogeosciences, 15, 2251–2269, https://doi.org/10.5194/bg-15-2251-2018, 2018.

[Figure]

**Figure R1** Nutrient limation factors for phytoplankton and benthic algae

**Second,** there is no mention of groundwater inputs of N or loss by hyporheic exchanges, when we are here in a context of a large lowland river where agriculture is likely important, i.e. conditions where diffuse sources of N in the intermediate catchment between the upstream and downstream boundaries can be significant. These sources can be particularly significant during summer low flows, if the geology near the river allows for it. Given the numbers on how little N is being transformed within the river corridor studied here (compared to the loadings), I question the certainty of the results since N diffuse sources are not accounted for. Could it be that these are of a similar order of magnitude as the reactive processes as the water moves downstream? I'd like this to be discussed somewhere in the Discussion.

**Responses:**

According to the discharge comparison at the upstream and downstream stations, they show very close values (Figure R2). This demonstrates the limited contribution fraction of the lateral inputs in the study reach. There are two potential N diffuse sources, namely tributaries and direct groundwater inputs. In the manuscript, we have considered the lateral input from the tributaries. According to our water balance calculation results over the five years, we found that the mean percentage error for 5 years was +0.97%[2] (in the SI). Targeted on the low flow period, we calculated the water balance taking the example of the extreme summer low flow period in 2018. Here we have got the result of the imbalance percentage is
* * *
[2] The positive value means that discharge at the outlet was lower than the input.

+0.59%. These small percentage values both on a multiannual basis and for extreme summer low flow period suggest the direct evidence of that groundwater flow exchange in the study reach does not play a significant role in the water balance. In fact, the 8 small tributaries are used as drainage of the corresponding sub-catchments. The flow and N concentration at the outlet of the tributaries are the results of hydrological cycle including the groundwater exchanges in the sub-catchments. The direct exchange with groundwater of the main stem of the study reach in the Lower Bode is therefore very limited.

In the summer low flow periods, the uncertainty of uptake estimation caused by the lateral boundary conditions are small because of three reasons. First, most inflow and loading come from the upstream boundary (Figure R2), which is well constrained by the high-frequency discharge and nitrate measurements. Second, the water imbalance is very small (0.59% in summer 2018), see also Figure R2. During the extreme low flow conditions in 2018 the tributaries were all dry as documented by personal visual inspections and the state water authority, which means errors of calculating these discharge inputs can be neglected. If we further assume that at extreme low flows lateral inflows should be highest our well-balanced discharge during this extreme discharge conditions suggest that there are no significant inflows not only during this low flow conditions but also during higher flows stages. Third, the net uptake percentage of the total input loadings is high in this season (nearly 30% in summer 2018). As the reviewer suggested, the uncertainty in instream N uptake estimation caused by the uncertainty in diffuse sources will be supplemented to the discussion in the manuscript.

[Figure]

**Figure R2** Discharge comparison at GGL and STF

**Finally,** I was wondering how the model deals with data inputs of different temporal frequencies. For instance, if Q, NO3, CHLa or DO are measured every 15min and serve as model forcing, were $PO_4^{3-}$, $NH_4^+$, and other variables measured at a low frequency interpolated before being used as data inputs? This could be a critical point, in particular for P since it is such an important variable controlling phytoplankton dynamics.

**Responses:**

This is a good question from an experienced modeler. Although this study has a very good dataset with paired high-frequency measurements for many variables, still some water quality variables, e.g., $PO_4^{3-}$, $NH_4^+$, and CBOD, are not yet available at high frequency. It could still be the case for many studies dealing with water quality modelling with high-frequency measurement data. The model we use here offers two options for input data interpolation, namely linear and step interpolation. In our case, we used linear interpolation. We took this assumption based on the consistent observed seasonal patterns with multi-annual records at GGL and STF for the upper boundary condition (Figure R3).

[Figure]

**Figure R3** PO$_4^{3-}$ concentration at GGL from 2007 to 2018

There could be discontinuities of P concentrations (either shown as enrichment effect or dilution effect) falling out of the linear interpolation line of monthly measurements, which is most probably caused by precipitation events. As P availability is a limiting factor for the phytoplankton and benthic algae growth (in Figure R1), such changes can reduce or increase the growth limitation of algae in the short term, thus affecting their growth rate, which in turn affects the N assimilatory uptake rate. This effect might be stronger for phytoplankton because benthic algae have internal phosphorus storage in their biomass, which can adjust and buffer the response to the sudden change of the P availability in the environment.

Our model was able to nicely capture the phytoplankton growth between GGL and STF for several years which we could prove with our high frequency Chl-a measurements. Hence, we can assume that we also captured the impact of PO$_4^{3-}$ on algae growth reasonably. The same is true for GPP calculations which represents the sum of phytoplankton and benthic algae metabolism and can be proved by high frequency DO data. In addition, long term PO$_4^{3-}$ data suggest a robust seasonal variation in PO$_4^{3-}$ and that this variation is much larger than short term PO$_4^{3-}$ variation. This is also confirmed by our modelling results which show relatively small short-term, e.g., diurnal, variation of PO$_4^{3-}$ (Figure R4). Furthermore, we can assume that PO$_4^{3-}$ variation is highly discharge dependent. This means that during stable low flow conditions also PO$_4^{3-}$ concentration do not vary much under such conditions. This may explain why infrequent PO$_4^{3-}$ measurements allowed us to nicely capture algae growth. One has to keep in mind that highest phytoplankton growth is mostly associated during stable low flow conditions and the absence of discharge disturbances. Taking this into account it appears that our infrequent PO$_4^{3-}$ concentration measurements still allowed a reasonable algae and N uptake calculations. Nevertheless, there might be room to improve especially phytoplankton simulations and N uptake with high-frequency PO$_4^{3-}$ measurements. Ongoing work investigates short term variation of PO$_4^{3-}$ using the Seabird P analyser which offers 30-minute measurement frequencies to better link PO$_4^{3-}$ and N uptake

by primary production. The above discussion will be supplemented in the discussion section as the reviewer suggested.

I raised some other important elements, as detailed below, and some minor technical corrections that need to be integrated too.

**Major issues, questions or comments**

**L10:** why is this so urgently needed? Please provide in half a sentence a bit more of context on N in large rivers in an agricultural context.

**Response:** Thanks for the comment. We will provide the relevant content and replace the first sentence to the sentences below in the revised version. "Large agricultural streams receive excessive inputs of nitrogen. However, quantifying the role of large agricultural streams in processing nitrogen remains limited because continuous direct measurements of complex interacting and highly time-varying nitrogen processing pathways in larger streams and rivers are difficult."

**L51:** quid of macrophytes versus periphyton contributions? Macrophytes are increasingly important in some large rivers (Seine, Moselle, Loire, Ebre), because of invasive species, and should certainly be mentioned too.

Some references on this topic that could be considered here and elsewhere in the manuscript when appropriate:

Flipo, N., Even, S., Poulin, M., Tusseau-Vuillemin, M.-H., Ameziane, T. and Dauta, A.: Biogeochemical modelling at the river scale: plankton and periphyton dynamics, Ecol. Modell., 176(3–4), 333–347, doi:10.1016/j.ecolmodel.2004.01.012, 2004.

Desmet, N. J. S., Van Belleghem, S., Seuntjens, P., Bouma, T. J., Buis, K. and Meire, P.: Quantification of the impact of macrophytes on oxygen dynamics and nitrogen retention in a vegetated lowland river, Phys. Chem. Earth, Parts A/B/C, 36(12), 479–489, doi:10.1016/j.pce.2008.06.002, 2011.

Hilton, J., O'Hare, M., Bowes, M. J. and Jones, J. I.: How green is my river? A new paradigm of eutrophication in rivers., Sci. Total Environ., 365(1–3), 66–83, doi:10.1016/j.scitotenv.2006.02.055, 2006.

Ibanez, C., Prat, N., Duran, C., Pardos, M., Munné, A., Andreu, R., Caiola, N., Cid, N., Hampel, H., Sanchez, R. and Trobajo, R.: Changes in dissolved nutrients in the lower Ebro river: Causes and consequences, Limnetica, 27(1), 131–142, 2008.

Minaudo, C., Abonyi, A., Leitão, M., Lançon, A. M., Floury, M., Descy, J.-P. and Moatar, F.: Long-term impacts of nutrient control, climate change, and invasive clams on phytoplankton and

cyanobacteria biomass in a large temperate river, Sci. Total Environ., 756, 144074, doi:10.1016/j.scitotenv.2020.144074, 2021.

Diamond, J. S., Moatar, F., Cohen, M. J., Poirel, A., Martinet, C., Maire, A. and Pinay, G.: Metabolic regime shifts and ecosystem state changes are decoupled in a large river, Limnol. Oceanogr., lno.11789, doi:10.1002/lno.11789, 2021.

**Response:** Thanks for these valuable literature for more information about quid of macrophytes versus periphyton contributions. We will cite some of them in the revised version. We will also mention the increasing importance of Macrophytes in some European large rivers as the reviewer suggested.

**L78-79:** DIN uptake is one of the possible pathways. Put like this it clearly insists on biological uptake, and banalizes the other processes. Please consider changing to "how temporally variable are the DIN pathways on a daily scale?"

**Response:** We agree with the reviewer's suggestion to change it to "how temporally variable are the DIN pathways on a daily scale?"

**L98-99:** "The mean depth of the reach is 60 cm. The mean stream width is 20 m.". How and when was this obtained? Are these annual means? Certainly, that a sense of seasonality could be of additional information: how deep does it get in summer lowflow compared to winter?

**Response:** In the manuscript, the information was obtained from an expert estimate. For describing the Lower Bode more accurately, we have calculated the annual mean river depth from the hydraulic model during the study period for the study reach to be approximately 1 m. The annual mean stream width is 20 m in the study period for the study reach. In the summer low flow period, the river depth is about 0.5 m low. However, the depth during the winter high flow can be as high as 2-3 meters. This information will be supplemented in the revised version.

**L103:** no signs of macrophytes? Or is this included in "benthic algae"? Please provide more information.

**Response:** The reviewer has answered this question in the latter comment by himself. We mentioned in Line 342 that we did not distinguish periphyton and macrophytes for benthic algae in our modelling practice. The relevant information will also be supplemented in the method section instead of only in the discussion as suggested by the reviewer.

**L123:** were these regressions always of good quality?

**Response:** The $R^2$ for correlation of Q and $NO_3^-$ is given in Table S1. The correlation for tributary Sarre was strong. The correlations were moderate for Ehle and Hecklinger Hauptgraben and weak for Sülzgraben and Marbegraben. Only the correlation at the tributary Beek was very weak.

**L151:** Are they each of them of 806 m length, or is this the average? If it's in an average, how was this segmentation defined?

**Response:** 806 m is an average value of the length of the 34 segments. The segments were defined considering cross-sectional profiles, tributary outlet locations, and spatial precision. The cross-sectional profiles within one segment should have no abrupt change. The tributaries are discharged to an individual segment. And the defined segmentation is able to reproduces the observe concentration differences between the upstream and downstream stations.

**L154:** were these variables interpolated at a higher frequency? How does the model accept inputs of different temporal frequency? If there were some interpolation involved, how was this done exactly, because all these 3 variables are susceptible to diel cycles.

**Response:** Linear interpolations of monthly $NH_4^+$, $PO_4^{3-}$, and TP were done in certain routine embedded in the WASP program, which is able to interpolate the input data of different temporal frequencies to fit the computation time step. The main justification of the linear interpolation is the consistent seasonal pattern shown by a decade of monthly observations. We agree with the reviewer on that the 3 variables are susceptible to diel cycles. Even with linear interpolation of the upstream boundary condition from monthly data, we can observe the diurnal patterns of the ammonium and phosphorus computed from the WASP model (Figure R4), especially during the period when their concentrations are low. Nevertheless, the magnitude of diurnal variations caused by the instream processes does not overshoot the seasonal pattern (Figure R4). Therefore, by linearly interpolating the monthly data which might contain diel patterns, we may bring uncertainty in estimating reach-scale N uptake on a daily scale in a small magnitude. However, it won't change the overall results of estimation. Some more discussion related to $PO_4^{3-}$ was given above in the response to the reviewer's third important question as well. We will supplement the relevant discussion in the revised version.

[Figure]

**Figure R4** $PO_4^{3-}$ and $NH_4^+$ concentrations from 2014 to 2018 and zoom-in July to September 2018

**L154:** Groundwater inputs/outputs are not included in the model? Please comment on this aspect, since it can be an important source or sink, especially significant during summer low flows.

**Response:** This aspect has been explained in the response to the reviewer's second important question.

**L163:** "and 2 additional parameters sensitive to DO and Chl-a were identified" which ones?

**Response:** Temperature coefficient for benthic algal nutrient excretion ($\theta_{Eb20}$) and phytoplankton maximum growth rate constant at 20 °C ($k_{Gmax}$) are the 2 additional parameters which are sensitive to DO and Chl-a simulations but out of the top 10 ranking of sensitivity parameters for $NH_4^+$ and $NO_3^-$.

**L177:** is ROC a constant or is this time variant based on phytoplankton communities? Please detail

**Response:** Yes, ROC is a constant. The value is 2.67 for both benthic algae and phytoplankton. This value was missing in the manuscript, and we will supplement it in Table S2.

**L178:** same as for ROC, is ADC a constant? Please detail

**Response:** Yes, ADC is a constant. The value is 2.5 for both benthic algae. This value was missing in the manuscript, and we will also supplement it in Table S2.

**L222:** Figure 2. Although I understand P is not the centre of attention here, why is it not represented? I'm guessing that just like in most European rivers, P is limiting factor for river primary production, and having a look at how good the model performs for it and how it behaves seasonally would be useful.

Please consider adding PO4 in the manuscript, not only in SI.

Also, I'm wondering how much the signal at the outlet (STF) differs from the signal at the reach input GGL. There are good chances that in terms of concentrations, in and outputs are pretty close, except for CHLa, DO, and PO4 when GPP gets really significant. I think the reader needs to visualize it, it could be done by adding the timeseries for GGL in the plots from Figure 2.

**Response:** We will put the $PO_4^{3-}$ plot in the manuscript together in Figure 2 as the reviewer suggested. As the reviewer mentioned here, P is the limiting factor for river primary production, and having a look at how well the model performs for it and how it behaves seasonally would be helpful. More explanations are also given in the response to the reviewer's first important question.

That is a very good comment. We fully agree that the readers need to see the comparison of upstream and downstream concentrations for their understanding. As integration of the time series for GGL in the plots in Figure 2 disturbs the comparison of simulation and validation at STF, the comparison plot will be put into another figure in the SI.

**L295:** I also think that travel time is essential for primary production to occur in the river. Please add this essential component in this sentence too.

**Response:** Absolute. The travel time at high flow is shorter. We will add the relevant phrasing in this sentence. The time-series plot for travel time can also be supplemented in the SI.

**L304:** Can you explain why it is critical?

**Response:** Yes, we may overvalue the role of streams in N uptake if we don't consider the N release phenomena. We will supplement an adverbial clause of reason in the current sentence to explain it.

**L305:** the performance of what? of the model?

**Response:** Here we meant the performance of study reach in processing DIN. We will supplement this information in the manuscript to avoid misunderstanding.

**L310:** "might cause significant uncertainties in estimating the role of streams in annual DIN uptake": in which sense? We likely overestimate annual net DIN uptake if we only consider measurements taken un summer. Please make it clearer.

**Response:** In the growing season, the net N uptake percentage can be as high as almost 30%. By only looking at the results from the experiments in this season, we might have an impression that the river is very efficient in removing N generally. However, the input N loadings of the growing season (spring + summer) accounted for only 37% of the total input loadings for the whole year (calculated from Table 4). In the lower Bode, the concentration of N and discharge is positively correlated, which can be observed in many other agriculture streams in Europe. As the Lower Bode receives higher N input loadings and becomes less efficient in removing N in the non-growing season, its average percentage of the net N uptake on a yearly basis was only 1.2%. We will supplement the explanations in the manuscript as the reviewer suggested.

**L336-337:** quid of phosphorus limitation? Could it be that once P resources are depleted, phytoplankton biomass collapses and this profits to benthic algae which needs lower nutrients or can take it from the sediment, impeding another seasonal bloom of phytoplankton? I think more detail on the origin and fate of P in the river reach considered in this study is needed.

**Response:** The disappearance of the spring phytoplankton peak in the study reach is mainly driven by the decreasing phytoplankton concentration at the upstream boundaries as seeds (Figure R5). The seasonal pattern of Chl-a can be intensified for example by further accumulation of the phytoplankton within the reach in the spring bloom. However, the start and disappear of the spring bloom peak is synchronized at GGL and STF. As shown in Figure R6, the $PO_4^{3-}$ concertation was the lowest around the spring phytoplankton bloom peak and increased until the summer, reaching the highest in June to July. This suggests that the absence of phytoplankton bloom in summer is unlikely due to the P limitation. Therefore, the P availability is unlikely to be the controlling factor in the transition of spring phytoplankton bloom to summer benthic algae dominance. The seasonal patterns of both Chl-a and

$PO_4^{3-}$ concentrations have been already shaped in the upstream course of the study reach. More investigation is needed to determine the reason for the seasonal shift of phytoplankton and benthic algae dominance, considering the development of phytoplankton in the course of the whole Bode. More discussion on P will be supplemented, as also mentioned above, in the responses to the reviewer's first and third important questions.

[Figure]

**Figure R5** Chl-a concentrations at GGL and STF (simulated) during the phytoplankton bloom from April to May 2017

[Figure]

**Figure R6** Simulated Chl-a and $PO_4^{3-}$ concentrations at STF

**L339:** are these rivers of similar geomorphological context and anthropogenic pressures?

**Response:** Not exactly. The Nemunas River is the largest river in Lithuania with the depth from 1.5 to 5 m with the bed width from 80 m to 200-300 m. The Nemunas flows at about 1 to 2 m/s. The depth is similar to the Lower Bode, but the width is much larger than that of the Lower Bode. In the literatures that the reviewer recommended to us above, we also found seasonal patterns of benthic algae in the agricultural rivers with a similar geomorphological context, which will be cited here as well.

**L340-341:** is grazing so important in the river Bode? Please provide more information, since grazing is usually a negligible sink term for phytoplankton.

If it is not so important, then increasing T°C, decreasing turbulence, higher irradiance, longer travel time should enable blooms of chlorophytes. How do you explain this is not the case?

**Response:** Sorry for the misunderstanding. We confirm the reviewer's judgment that grazing is a negligible sink term for phytoplankton in the Lower Bode. This sentence in the current version of manuscript is misleading the reader. We will rephrase the saying about it to avoid misunderstanding.

As written in the response to L336-337 above, the most decisive factor of the phytoplankton bloom in the study reach is in fact the Chl-a concentration at the upstream station GGL (Figure R5). It basically shapes the seasonal pattern and the bloom peaking of Chl-a concentration in the whole study reach (Figure R5). Reynolds and Descy (1996)[3] mentioned in the classic review paper about their three provenances on the phytoplankton in large rivers, namely wash-off from benthic epiliths and epiphytes, the presence in lakes and impoundments along the course of the river or its tributary streams and the ability to survive within a unilateral flow moving everything fatally seawards. The provenance of phytoplankton upstream of the GGL is difficult to say and out of the scope of this study. We have mentioned the possible mechanism of grazing here because we know that there are some impoundments upstream of the Lower Bode reach. For standing waters like lakes and reservoirs, Sommer et al. (2012)[4] suggest that the spring phytoplankton peak is believed to be suppressed by the increasing grazing zooplankton. To avoid misunderstanding, we will clarify in the manuscript the dominant controller of the phytoplankton bloom in the lower Bode reach is the Chl-a concentration from the upstream and the instream environment conditions.
* * *
[3] Reynolds, C.S. and Descy, J.P.: The production, biomass and structure of phytoplankton in large rivers, Large Rivers, 10, 161-187, https://doi.org/10.1127/lr/10/1996/161, 1996.

[4] Sommer, U., Adrian, R., De Senerpont Domis, L., Elser, J.J., Gaedke, U., Ibelings, B., Jeppesen, E., Lürling, M., Molinero, J.C., Mooij, W.M., van Donk, E., and Winder, M.: Beyond the Plankton Ecology Group (PEG) Model: Mechanisms Driving Plankton Succession, Annu. Rev. Ecol. Evol., 43(1), 429-448, https://doi.org/10.1146/annurev-ecolsys-110411-160251, 2012.

As for the second comment, the other factors like decreasing turbulence, higher irradiance, longer travel time in summer can promote the phytoplankton development in the study reach. Still the upstream seed concentration is most decisive. Increasing water temperature in summer does not necessarily promote the blooms as diatoms, which are the predominant species in Lower Bode, grow optimally at lower temperatures.

**L342**: This answers some previous comments I raised. Please explain this earlier in the Method section

**Response:** Yes, we will explain this earlier in the method section. Thank you!

**L347:** what are you referring to? Please provide more info what you call "characteristic time"

**Response:** Characteristic time is simply a measure of how fast a process will proceed. The process refers to that plants act as a biological pump incorporating the pelagic N into the benthic compartment cited from Lepoint et al. (2004)[5]. During the phytoplankton blooms in spring, large amounts of N are taken up over several days and weeks. In contrast, the uptake of N by benthic algae was slower and spread out over several months of the year.

**L351-352:** "the difficulties inherent in the use of high-frequency chlorophyll fluorescence signal as indirect measures of phytoplankton biomass". Please explain and discuss more on this, because it is important. Chlorophyll a was shown to be a poor proxy for phytoplankton biomass, because of dynamic chloroplast packaging in phytoplankton cells depending on their ecophysiology, because of changing phytoplankton species, … etc. Also, fluorescence is subject to large uncertainties if the measurements are done directly into the stream and not protected from solar irradiance, a phenomenon called non-photochemical quenching. Under large irradiance, CHLa can be underestimated by 50%. Is this the case? If yes, please make it clear and raise this as an element of discussion.

**Response:** In the Lower Bode case, we can even observe the diurnal pattern of Chl-a concentration in summer, which can be reproduced by the model (Figure R7). This phenomenon was also reported in a recent study Pathak et al. (2021)[6] in an agricultural lower land river in the UK. In these two cases, there is no evidence of the impact of non-photochemical quenching on Chl-a fluorescence signal in summer

[5] Lepoint, G., Gobert, S., Dauby, P., and Bouquegneau, J.-M.: Contributions of benthic and planktonic primary producers to nitrate and ammonium uptake fluxes in a nutrient-poor shallow coastal area (Corsica, NW Mediterranean), J. Exp. Mar. Biol. Ecol., 302(1), 107-122, https://doi.org/10.1016/j.jembe.2003.10.005, 2004.

[6] Pathak, D., Hutchins, M., Brown, L., Loewenthal, M., Scarlett, P., Armstrong, L., Nicholls, D., Bowes, M., and Edwards, F..: Hourly prediction of phytoplankton biomass and its environmental controls in lowland rivers, Water Resources Research, 57, e2020WR028773. https://doi.org/10.1029/2020WR028773, 2021

as shown in the high-frequency measurements for Lake Rotoehu shown in Hamilton et al. (2015)[7]. This information will be discussed in the revised version, as the reviewer suggested.

[Figure]

**Figure R7** Simulated Chl-a concentrations at STF in July and August 2018

**L355:** please explain what kind of disturbance

**Response:** Here we refer to temporal disturbances caused by runoff events, such as reduced light availability during the phytoplankton bloom and detachment from the river bottom in high flow events. We will specify the disturbances in the revised manuscript.

**L383:** yes, but is it is also likely that reactive P sources are from point sources, and therefore would overall be diluted during high flows rather than transported. Please provide more information on this particularly important aspect: P resources are often scarce during the blooming season, might get depleted, and constrain the entire algal biomass modelling exercise

**Response:** Yes, it is likely that the main reactive P sources are from point sources in the Lower Bode according to its C-Q relationship. We meant here only that P is likely to be released from the plants as the N element does. Even with the release loads, the instream P concentration could also be shown with a dilution effect during high flows. It is not contradictory. Thanks a lot for raising this particularly important aspect in P. As already mentioned in several responses to the reviewer's questions, we will supplement more discussion about P in the manuscript.

**Technical corrections**

**L66:** Another example of a biogeochemical modelling approach in a large river is Minaudo, C., Curie, F., Jullian, Y., Gassama, N. and Moatar, F.: QUAL-NET, a high temporal-
* * *
[7] Hamilton, D.P., Carey, C.C., Arvola, L., Arzberger, P., Brewer, C., Cole, J.J., Gaiser, E., Hanson, P.C., Ibelings, B.W., Jennings, E., Kratz, T.K., Lin, F.-P., McBride, C.G., David de Marques, M., Muraoka, K., Nishri, A., Qin, B., Read, J.S., Rose, K.C., Ryder, E., Weathers, K.C., Zhu, G., Trolle, D., and Brookes, J.D.: A Global Lake Ecological Observatory Network (GLEON) for synthesising high-frequency sensor data for validation of deterministic ecological models, Inland Waters, 5(1), 49-56, https://doi.org/10.5268/IW-5.1.566, 2015.

resolution eutrophication model for large hydrographic networks, Biogeosciences, 15(7), 2251–2269, doi:10.5194/bg-15-2251-2018, 2018.

**Response:** Thanks a lot for mentioning this. We have read it and found it very relevant! We will compare our study results with this study in many aspects, e.g., the value of high-frequency data in water quality modelling, the role of P, etc. It will be cited in the manuscript.

**L90:** "the lower reaches ARE dominated"

**Response:** We will correct it accordingly.

**L94:** Is "donate" the proper word?

Pink and red can be easily confused. Since circle sizes are different, please change this part of the caption to "The small pink circles" or use another marker type (square, triangle...)

**Response:** It was a typo. We will change it to represent. Also, thank you for the practical suggestions about the styling. We will revise the marker type and colour in Figure 1 accordingly.

**L96:** "respectively. The grey shaded area represents the Selke sub-catchment" why is this important?

**Response:** The 4th order Selke is mentioned in Line 316-318 in the relevant comparison and discussion on benthic algae and phytoplankton uptake pathways.

L124: Please delete "Meanwhile,"

**Response:** Thanks! We will delete it.

L145: Table 1: All these factors 1000 could be avoided simply by specifying that units are in gN/m2/d instead of mgN/m2/d

**Response:** This is a good idea. We will specify the unit in $mgN/m^2/d$ for the table and remove all these factors 1000. In the footnote of the table, we will mention this unit conversion because in the manuscript we use the unit of $mgN/m^2/d$.

**L147:** "algal cell N in mgN/gD". What is the D in gD referring to? Please explain these units. I found out later it is related to "detritus", though it's clearly not an obvious notation.

**Response:** Thanks! Yes, you are right about detritus. We will add the explanation directly after "algal cell N in mgN/gD".

**L171:** please add you expressed GPP in g O2/m2/d

**Response:** Thank you for this suggestion! We will add it.

**L190:** shouldn't it be U_MIN instead of U_MIM?

**Response:** Yes, it should be U_MIN. We will change it to U_MIN.

**L272:** "from the perspective of DIN" is vague, please revise this sentence

**Response:** Thanks for the suggestion. We will change the phrase to be "One explanation is that the net uptake was calculated only for DIN".

**L301:** "N has a round-trip ticket to the benthic algae": Please revise and adopt a more formal description.

**Response:** "N has a round-trip ticket" was actually a saying from von Schiller et al. (2015)[8]. We used it here because it is very illustrative way to describe N uptake and release to and from benthic algae. We will keep the saying but use a quotation mark for "round-trip ticket".

**L306:** "despite the highest percentage being close to 30%.": at the daily scale, right? Please revise

**Response:** Yes, thanks! We will change it to "despite the highest percentage being close to 30% at the daily scale".

**L308:** "Moreover, there is also a seasonal shift to net release in an annual cycle." This sentence is unclear, please revise.

**Response:** We will change this sentence to "Moreover, there is also a seasonal shift from net uptake to net release in an annual cycle."

**L325-326:** "along with": Please revise

**Response:** We will change "along with" to "across".

**L445:** References: There are some suspicious doi links in the references, please revise in particular the ones below:

Burgin, A.J. and Hamilton, S.K.: Have we overemphasized the role of denitrification in aquatic ecosystems? A review of nitrate removal pathways, Front. Ecol. Environ., 5(2), 89-96, https://doi.org/10.1890/1540-9295(2007)5[89:HWOTRO]2.0.CO;2, 2007.

Rutherford, J.C., Young, R.G., Quinn, J.M., Chapra, S.C., and Wilcock, R.J.: Nutrient attenuation in streams: a simplified model to explain field observations, J. Environ. Eng., 146(8): 04020092, https://doi.org/10.1061/(ASCE)EE.1943-7870.0001753, 2020.

Tank, J.L., Reisinger, A.J., and Rosi, E.J.: Chapter 31 - Nutrient limitation and uptake, in: Methods in Stream Ecology (Third Edition), edited by: Lamberti, G.A. and and Hauer, F.R., Academic Press, Elsevier, 147-171, https://doi.org/10.1016/B978-0-12-813047-6.00009-7, 2017.
* * *
[8] von Schiller, D., Bernal, S., Sabater, F., and Martí, E.: A round-trip ticket: the importance of release processes for in-stream nutrient spiraling, Freshw. Sci., 34(1), 20-30, https://doi.org/10.1086/679015, 2015.

**Response:** Thanks! The doi links look suspicious, but we have checked them all one by one. They are all correct.

Once again, we appreciate the critical comments and constructive suggestions from the reviewer very much.

Jingshui Huang

Contact author behalf of all co-authors